# `DiffCATS`: **Causally Associated Time-Series Generation through Diffusion Models**

**Giuseppe Masi** *masi.g@di.uniroma1.it*
*Sapienza University of Rome*

**Andrea Coletta**[*] *andrea.coletta@bancaditalia.it*
*Banca d'Italia*

**Elizabeth Fons** *elizabeth.fons@jpmorgan.com*
*J.P. Morgan AI Research*

**Svitlana Vyetrenko** *svitlana@outsampler.com*
*Outsampler, Strasbourg, France*

**Novella Bartolini** *novella@di.uniroma1.it*
*Sapienza University of Rome*

**Reviewed on OpenReview:** *https://openreview.net/forum?id=FwC6CyaHop*

## Abstract

Modeling and recovering causal relationships in time-series data can be crucial for supporting real-world interventions and decision-making, but progress in Time-Series Causal Discovery (TSCD) is often limited by the lack of high-quality datasets with diverse and realistic temporal causal relationships. This highlights the need to provide synthetic time-series generation tools, with realism as a primary objective, an aspect that requires incorporating causal relationships beyond mere correlation. To address this challenge, we propose a diffusion model called `DiffCATS`. It simultaneously generates multiple causally associated time-series as well as a ground truth causal graph that reflects their mutual temporal dependencies, requiring only observational time-series data for training. Experiments demonstrate that it outperforms state-of-the-art methods in producing realistic time-series with causal graphs that closely resemble those of real-world phenomena. We highlight the practical utility of our data on three downstream tasks, including benchmarking widely used TSCD algorithms.

## 1 Introduction

Many real-world time-series can be usefully modeled as arising from directed (causal) relationships among variables, which motivates methods for representing and learning such structure from data (Runge, 2018; Runge et al., 2023). Understanding such causal relationships is a well-recognized and important challenge for decision-making and policy formulation, as it facilitates predicting the consequences of interventions on underlying systems and variables (Hasan et al., 2023).

Over the years, several works have studied these underlying causal structures, starting from Granger Causality (GC) (Granger, 1969). Unable to capture how time affects causal relationships between interdependent time-series, GC has been complemented by Causal Graphs that incorporate the temporal lag in which causality unfolds (Pearl, 2009). Many approaches tackling the Time-Series Causal Discovery (TSCD) problem (Hasan et al., 2023) achieve satisfactory performance using statistical and machine learning techniques (Runge

---

[*]The opinions expressed in this paper are personal, and should not be attributed to Banca d'Italia. The research work was carried out before the author joined Banca d'Italia.

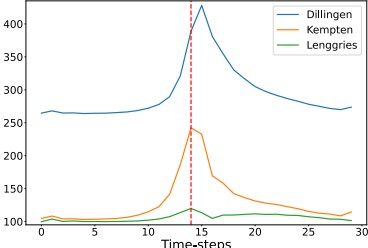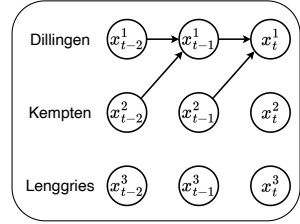

Figure 1: A synthetic sample and causal graph of three river discharges in which Kempten has an effect on Dillingen with a lag of 1.

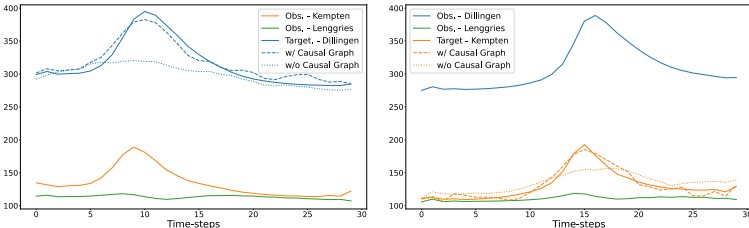

Figure 2: Two generated examples in which the causal graph helps a predictor model to reconstruct the target feature.

et al., 2019; Pamfil et al., 2020; Sun et al., 2023; Cheng et al., 2023), with discovered causal graphs closely resembling the ground-truth counterparts. However, the limited data available may hinder the development of new methodologies and studies, raising concerns about how existing algorithms would perform in unseen real-world scenarios (Cheng et al., 2024).

Novel methodologies to generate realistic datasets with rigorously defined causal graphs are needed to support research and development of algorithms on time-series causal graphs. This challenge has been recently tackled by the works of Li et al. (2023) and Cheng et al. (2024), which mark an initial step in this direction, proposing two deep learning models to generate synthetic time-series data, while extracting the corresponding causal graphs. The first model focuses on the concept of Granger causality and proposes a recurrent Variational Autoencoder (CR-VAE) framework that naturally encodes causality into the weight matrix connecting input and hidden states. The second model introduces a comprehensive framework that supports prior causal graphs to generate realistic time-series data. However, when an input causal graph is not provided, the method extracts a hypothesized causal graph using explainability tools for feature importance (e.g., DeepSHAP (Lundberg, 2017)), which are inherently slow and only provide a posterior approximation of the ground-truth graph.

In this paper, we introduce a novel generative framework called **DiffCATS** that combines the advantages of previous approaches by jointly generating time-series along with its causal graph, directly within a diffusion model architecture. Specifically, our model incorporates a $\tau$-lag vector autoregressive structure (VAR($\tau$)) for multivariate time-series (Hamilton, 2020), where the coefficients are learned and generated through the diffusion process. This approach enables the simultaneous generation of realistic time-series data and the derivation of the corresponding ground-truth causal graphs from the VAR coefficients (Zivot & Wang, 2006). **DiffCATS** can be trained directly on time-series data without requiring prior causal graphs, eliminating the need for additional explainability tools.

Our work facilitates research and development of efficient algorithms for uncovering cause-effect relationships in multivariate time-series across diverse fields. In particular, the generated synthetic data can complement real-world data by providing known ground-truth with a diverse set of causal structures, which is especially valuable when real data are scarce or have limited ground-truth causal structures. Different from existing work, our approach specifically addresses the coherence between the synthetic sample and its corresponding causal graph (see Figure 1), resulting in more realistic and useful synthetic data. As the experiments highlight, the generated causal graphs are key to understanding the underlying dynamics of time-series, improving predicting capabilities (see Figure 2).

The main contributions of this work are the following:

- We present `DiffCATS`, a novel pipeline that employs a diffusion model to generate realistic time-series along with their related causal graphs.

- With extensive experiments, we demonstrate that our method outperforms existing approaches, in terms of synthetic time-series quality and fidelity of causal graphs to real-world phenomena.

- We conduct an evaluation of existing causal discovery algorithms using our synthetically generated datasets, highlighting the practical benefits of our data.

- We demonstrate the utility of the causal graph and its coherence with time-series data through two additional downstream tasks.

## 2 Related work

Several works have addressed the generation of synthetic time-series starting from real datasets (Yoon et al., 2019; Jarrett et al., 2021; Rasul et al., 2021). Some approaches have focused on specific aspects, such as the correlation dynamics among variables (Seyfi et al., 2022; Masi et al., 2023), user-specified constraints (Coletta et al., 2023), or interpretable generation methods (Yuan & Qiao, 2024; Fons et al., 2024). However, only a few works delve into the generation of time-series along with their causal structure (Li et al., 2023; Cheng et al., 2024).

The work of Li et al. (2023) proposed a VAE-based framework capable of learning Granger causal relationships from real multivariate time-series. This approach derives causal relationships from the weight matrix of the model connecting the input and hidden states, learning a unique Granger causality matrix from the data to which all generated samples adhere. A recent work of Cheng et al. (2024) proposed a pipeline to generate realistic time-series along with the causal graph. However, their framework does not output an interpretable-by-design time-series, but it performs the hypothetical causal graph inference through DeepSHAP (Sundararajan & Najmi, 2020) on the trained generative model, introducing a considerable time overhead.

Our goal is to further explore this area and address the gaps in the current literature. Specifically, we aim to provide a novel generative approach to simultaneously generate multiple causally associated time-series and their causal graphs, incorporating temporal lags. We strive to generate a unique causal graph for each synthetic sample, introducing greater variety in the data and providing a naturally interpretable architecture.

## 3 Problem Formulation

### 3.1 Background Knowledge

**Causal Discovery** The Causal Discovery task aims to identify cause-effect relationships among the variables of a $d$-variate time-series $\boldsymbol{x} = (\boldsymbol{x}^1, \ldots, \boldsymbol{x}^d)$. We say that $\boldsymbol{x}^i$ has an effect on (or causes) $\boldsymbol{x}^j$ if the two variables reflect a real phenomenon in which events reflected in the values of $\boldsymbol{x}^i$ affect $\boldsymbol{x}^j$. Trivially, the cause must precede the effect, so it is important to consider also the lag $\tau$ that elapses between observing the cause event on $\boldsymbol{x}^i$ and the effect event on $\boldsymbol{x}^j$. Causal Discovery algorithms are employed to observe real data and point out the existence of causal relationships according to which $\boldsymbol{x}^i$ causes $\boldsymbol{x}^j$, after $\tau$ time-steps, returning $(\boldsymbol{x}^i, \boldsymbol{x}^j, \tau)$.

**Causal Graphs** Causal relationships are often represented in the form of *Causal Graphs*. Let $\tau_{max} \in \mathbb{N}^+$ be the maximum number of discrete time-steps ($\delta t$) we are interested in to model the cause-effect phenomena of $\boldsymbol{x}$. We define a Causal Graph $G = (V, E)$ where the vertices $V$ represent the time-series variables for the various time-steps between $-\tau_{max}$ and 0, and the edges $E$ represent their causal relationships. In particular, an edge $(x_{t_1}^i, x_{t_2}^j) \in E$ indicates that the variable $x^i$ has a causal implication on the value of the variable $x^j$ with a lag of $t_2 - t_1$ time-steps (i.e., $x_{t_1}^i \Rightarrow x_{t_2}^j$). Formally, $V = \{x_{t-l}^i \mid 0 < i \leq d, 0 \leq l \leq \tau_{max}\}$ and $E = \{(x_{t_1}^i, x_{t_2}^j) \mid x^i \Rightarrow x^j \text{ with a lag of } t_2 - t_1 \geq 0\}$. Figure 1 illustrates a causal graph describing the

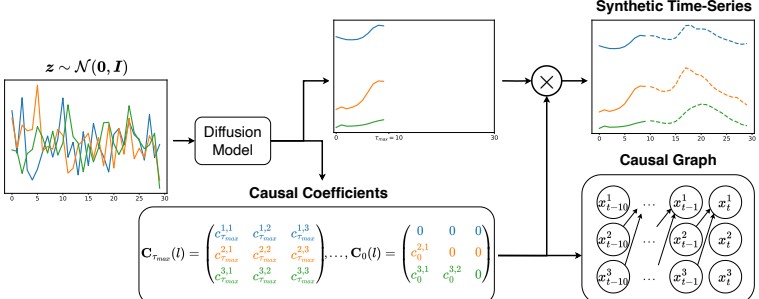

Figure 3: `DiffCATS` pipeline. Given a noisy sample $\boldsymbol{z} \sim \mathcal{N}(\boldsymbol{0}, \boldsymbol{I})$ the diffusion model runs the denoising steps and outputs (i) an initial prefix of $\tau_{max}$ steps of the multivariate time-series and (ii) causal/VAR coefficients $\{\boldsymbol{C}_\tau(l)\}_{\tau=0}^{\tau_{\max}}$, whose entries $c_\tau^{i_1,i_2}(l)$ parameterize directed influences from variable $i_1$ at lag $\tau$ to variable $i_2$ at time $l$. For each $l > \tau_{\max}$, the remaining samples are produced via causal reconstruction (circle "$\times$"), i.e., a VAR-style update using the previously reconstructed window and the generated matrices (including $\boldsymbol{C}_0(l)$). This yields the full synthetic time-series (solid prefix with dashed continuation). A sample-specific causal graph is then derived from the generated coefficients by retaining the most significant relationships, with nodes representing variables across time lags (e.g., $x_{t-10}^i, \ldots, x_t^i$).

interdependencies among rivers according to the observations made by Ahmad et al. (2022). It shows that variations in the water level of one river affect the level of the other one.

## 3.2 Task Definition

Let $\mathcal{D} = \{\boldsymbol{x} \mid \boldsymbol{x} \in \mathbb{R}^{L \times d}\}$ be a set of $d$-dimensional input time-series of length $L$. Our goal is to use the data in $\mathcal{D}$ to train a generative model that best approximates the distribution of the time-series, while simultaneously learning the corresponding causal structures. In particular, we aim at generating couples $\langle \hat{\boldsymbol{x}}, \hat{g} \rangle$ where $\hat{\boldsymbol{x}} \in \mathbb{R}^{L \times d}$ is a synthetic time-series and $\hat{g}$ is a causal graph. We want $\hat{\boldsymbol{x}}$ to be similar to the time-series observed in $\mathcal{D}$, i.e., to show a realistic behavior that reflects similar statistical properties. We also require $\hat{\boldsymbol{x}}$ to reproduce realistic causal relationships, as observed in $\mathcal{D}$, and described by the causal graph $\hat{g}$ that will explain $\hat{\boldsymbol{x}}$ in terms of causal relationships. We formally define a causal relationship between two time-series variables in the following section (Definition 1).

# 4 Methodology

Our framework, illustrated in Figure 3, builds upon a powerful class of generative models, the diffusion models (Ho et al., 2020). [1] Originally designed to generate realistic images, diffusion models are used in this paper to generate synthetic samples $\langle \hat{\boldsymbol{x}}, \hat{g} \rangle$. Unless otherwise noted, we adopt two common assumptions of the Causal Discovery literature (Cheng et al., 2024; Runge et al., 2019; Pamfil et al., 2020; Sun et al., 2023): *Markovian conditions* and *faithfulness*, as discussed in detail in Appendix A.1. While these assumptions are crucial for causal discovery, the key consideration for generation from observational data is ensuring structural consistency between the synthetic time-series and the causal graph. Indeed, unlike previous works, we do not need to assume *stationarity* of the underlying causal phenomena, as our causal graphs will be strictly related to individual samples.

## 4.1 Diffusion framework

A diffusion model is a type of latent variable model that operates through two key processes: the *forward process* and the *reverse process*. Given a sample $\boldsymbol{x}_0 \in \mathcal{D}^2$, the forward process gradually adds Gaussian noise

---

[1]Although the proposed methodology is general and can, in principle, be applied to alternative generative paradigms such as VAEs Kingma & Welling (2013) or GANs Goodfellow et al. (2014), these models often underperform diffusion models due to more restrictive assumptions and less stable training dynamics. Consequently, we focus on a diffusion-based architecture, which we carefully design for our setting.

[2]In this section, the subscript refers to the diffusion step, not the lag index used elsewhere.

to obtain a noisy sample $\boldsymbol{x}_T \sim \mathcal{N}(\mathbf{0}, \mathbf{I})$. Specifically, given the parameters $\beta_t \in (0, 1)$ to schedule the amount of noise added at diffusion step $t \in [1, T]$, the noisy sample $\boldsymbol{x}_t$ is given by $\boldsymbol{x}_t = \sqrt{\hat{\alpha}_t} \cdot \boldsymbol{x}_0 + \sqrt{1 - \hat{\alpha}_t} \cdot \boldsymbol{\epsilon}$ where $\boldsymbol{\epsilon} \sim \mathcal{N}(\mathbf{0}, \mathbf{I})$, $\alpha_t = 1 - \beta_t$, and $\hat{\alpha}_t = \prod_{i=1}^{t} \alpha_i$.

The reverse process performs the actual generation of a new sample starting from Gaussian noise. Following the formulation of Ho et al. (2020), we perform the denoising procedure from $\boldsymbol{x}_T \sim \mathcal{N}(\mathbf{0}, \mathbf{I})$ according to the following equation:

$$\boldsymbol{x}_{t-1} = \beta_t \cdot \frac{\sqrt{\hat{\alpha}_{t-1}}}{1 - \hat{\alpha}_t} \cdot \hat{\boldsymbol{x}}_0 + \frac{(1 - \hat{\alpha}_{t-1}) \cdot \sqrt{\alpha_t}}{1 - \hat{\alpha}_t} \cdot \boldsymbol{x}_t + \mathbb{1}_{\{t>0\}} \cdot \beta_t \cdot \frac{1 - \hat{\alpha}_{t-1}}{1 - \hat{\alpha}_t} \cdot \boldsymbol{\epsilon}. \tag{1}$$

In the above equation $\mathbb{1}_{\{\cdot\}}$ is the indicator function, $\boldsymbol{\epsilon} \sim \mathcal{N}(\mathbf{0}, \mathbf{I})$, and $\hat{\boldsymbol{x}}_0 = \text{DEN}_\theta(\boldsymbol{x}_t, t)$ is the output of a neural network $\text{DEN}_\theta$ parametrized by $\theta$, trained with respect the following loss function: $\mathcal{L}_{Rec}(\boldsymbol{x}_0, \hat{\boldsymbol{x}}_0; \theta) = \|\boldsymbol{x}_0 - \hat{\boldsymbol{x}}_0\|_2^2$, where $\|\cdot\|_p$ indicates the $\ell_p$-norm. In practice, $\text{DEN}_\theta$ reconstructs the original sample taken from the dataset by filtering out the noise added during the forward process.

In addition to the $\ell_2$-norm, we also consider other loss functions to improve the performance of the reconstruction, such as the Dynamic Time Warping-based term $\mathcal{L}_{DTW}(\boldsymbol{x}_0, \hat{\boldsymbol{x}}_0; \theta)$ introduced by Cuturi & Blondel (2017). The training objective is:

$$\mathcal{L}(\boldsymbol{x}_0, \hat{\boldsymbol{x}}_0; \theta) = \mathbb{E}_{\substack{t \sim \mathcal{U}(1,T) \\ \boldsymbol{x}_0 \sim \mathcal{D}}}[\mathcal{L}_{Rec}(\boldsymbol{x}_0, \hat{\boldsymbol{x}}_0; \theta) + \lambda_1 \cdot \mathcal{L}_{DTW}(\boldsymbol{x}_0, \hat{\boldsymbol{x}}_0; \theta)], \tag{2}$$

where $\lambda_1$ is a parameter weighting the additional term.

The architecture of $\text{DEN}_\theta$ consists of an initial convolutional layer followed by a series of RESNET and ATTENTION blocks (see Appendix B.3 for more details).

## 4.2 Causal reconstruction of the time-series

This section details how the output process inherently embeds a causal structure, allowing for the generation of a coherent sample $\langle \hat{\boldsymbol{x}}_0, \hat{g} \rangle$. Given $\boldsymbol{x}_0 \in \mathcal{D}$, we denote with $\boldsymbol{x}_0(l)$ the value of the time-series at time $l$, for $l \in [1, L]$, and considering its components as distinct features, we denote with $x_0^i(l)$ the value of the $i$-th feature at time $l$, for $i \in [1, d]$.

Let $\tau_{max} \in \mathbb{N}^+$ be the maximum lag for the causal relationships in the synthetic time-series[3]. Simultaneously for each feature $i$, $\text{DEN}_\theta$ outputs the first $\tau_{max}$ steps of the time-series, i.e., $\hat{x}_0^i(l)$, $\forall 1 \leq l \leq \tau_{max}$, and a set of causal matrices $\mathbf{C}(l) = \left\{ \mathbf{C}_\tau(l) = \left(c_\tau^{i_1, i_2}(l)\right)_{1 \leq i_1, i_2 \leq d} \in \mathbb{R}^{d \times d} \mid 0 \leq \tau \leq \tau_{max} \right\}$ for all the remaining steps $\tau_{max} < l \leq L$, where $c_\tau^{i_1, i_2}(l)$ represents the impact of feature $i_1$ on $i_2$ with a lag of $\tau$ at time $l$. The reconstruction of the whole time-series according to causal relationships evolves through a Vector Autoregressive (VAR) model (Zivot & Wang, 2006): proceeding one step $l$ at a time with $\tau_{max} < l \leq L$, $\hat{\boldsymbol{x}}(l) = \sum_{\tau=0}^{\tau_{max}} \mathbf{C}_\tau(l) \cdot \hat{\boldsymbol{x}}(l - \tau)$[4]

We underline that, even though the reconstruction can be described by a VAR model, the generation framework is not autoregressive. This is because the model does not consider previously generated outputs as inputs. It instead generates the initial time-steps and the coefficients simultaneously.

Finally, to encourage the model to focus on the most important causal relationships, we add a regularization term for the coefficients. While the ideal choice for such a function would be the $\ell_0$-norm, this is difficult to optimize, therefore we consider the $\ell_2$-norm, as in (Sun et al., 2023; Li et al., 2023). Specifically, the regularization is defined as:

$$\mathcal{L}_{Reg}(\hat{\boldsymbol{x}}_0; \theta) = \lambda_2 \cdot \sum_{l=\tau_{max}+1}^{L} \sum_{\tau=0}^{\tau_{max}} \|\mathbf{C}_\tau(l)\|_2, \tag{3}$$

where $\lambda_2$ is the weight associated to such regularization term.

---

[3]The maximum lag and time-step granularity can be decided according to the domain and expert knowledge.

[4]Note that the matrix modeling instantaneous relationships $\mathbf{C}_0(l), \forall l$ is generated as a lower triangular matrix with the elements on the main diagonal also set to 0, in agreement with Hyvärinen et al. (2010).

### 4.3 Causal Graph Extraction

While the broader field does not offer a single universally accepted characterization of causality in observational time-series, we will provide a flexible definition based on a predictive formulation that is both widely adopted in practice and grounded in robust theoretical principles. Indeed, given a synthetic sample $\hat{\boldsymbol{x}}_0$ reconstructed through the series of coefficients matrices $\boldsymbol{c}(l)$ it means that for each time-step $\tau_{max} \leq l \leq L$, for each feature $1 \leq i \leq d$, we have importance weights $\mathbf{C}_1(l), \ldots, \mathbf{C}_{\tau_{max}}$ assigned to the previously generated time-steps, i.e. the window $[\hat{\boldsymbol{x}}_0(l - \tau_{max}), \ldots, \hat{\boldsymbol{x}}_0(l - 1)]$ and instantaneous relationships in $\hat{\boldsymbol{x}}_0(l)$ according to $\mathbf{C}_0(l)$. We also call these coefficients the *explanation* of the synthetic sample. To infer the causal graph $\hat{g}$, we summarize the causal relationships from the VAR coefficients according to the following formal definition.

**Definition 1** *Let $\rho$ be the percentage of causal relationships we want to represent in the synthetic dataset. For a synthetic sample $\hat{\boldsymbol{x}}$, we say that $\hat{\boldsymbol{x}}^i \langle \rho, q \rangle$-causes $\hat{\boldsymbol{x}}^j$ with a lag of $\tau$ if the $q$-quantile of the corresponding coefficients of the VAR model at lag $\tau$ is among the $\rho\%$ highest absolute values. Notice that $\rho$ and $q$ refer to the whole dataset and the single sample, respectively.*

Intuitively, the $q$-quantile value allows the identification of significant causal events within a sample (e.g., Figure 1) by aggregating the coefficients over the time-series. Additionally, the threshold $\rho$ can be employed as a reference point to identify and constrain causality to focus on the most relevant causal relationships. The whole mechanism allows samples to have different causal graphs, i.e., some may exhibit dense connections while others may have none. In fact, unlike previous work, we do not assume stationarity of the causal relationships in the dataset: the causal graphs are strictly related to individual samples, enabling more realistic and diverse samples.

This approach enables a fair evaluation of TSCD algorithms (see Section 6.1). Specifically, a TSCD algorithm may fail to recover the ground-truth causal graph — derived from our prior knowledge of the system — if the given sample does not exhibit any causal effect but is still linked to that prior causal graph. Our dataset serves as an accurate and representative benchmark: each generated sample is explicitly linked to its corresponding causal graph, which may denote the absence of causal relationships when none emerge from the time-series.

Finally, we notice that our definition ensures that each identified causal link captures not only statistical predictability but also a meaningful, information-theoretic causal effect. In fact, our formulation is consistent with prior work that extends Granger causality with lag-specific formulations (e.g., (Hyvärinen et al., 2010)), as a significant predictive link also implies high transfer entropy from the cause to the effect (Runge et al., 2012), indicating a substantial information flow between two variables. This information flow is the type of relationship captured by our definition of causal links as discussed in more details in Appendix A.2.

## 5 Experiments

In the experiments section, we show that the proposed pipeline is able to generate high-quality synthetic samples along with coherent and realistic causal graphs. In this regard, we conducted an experimental campaign involving three different datasets. We compared our model against several state-of-the-art approaches to highlight its advantages. We evaluate the generated samples both quantitatively and qualitatively, using well-established metrics for synthetic time-series as well as metrics specifically designed to assess the realism of the causal graphs. The code to reproduce the experiments is publicly released[5].

### 5.1 Datasets

To evaluate the models' capability to generate time-series alongside their causal relationships, we utilize two real-world datasets (Rivers and AQI) and a synthetic dataset (Hénon) constructed using closed-form equations, for which we have ground-truth knowledge. The datasets are described in the following paragraphs while we refer the reader to Appendix B.1 for additional details, including the visualization of the ground-truth causal graphs.
- **Rivers**: introduced by Ahmad et al. (2022), it consists of the average daily ($\delta t = 1$ day) discharges of the

---

[5]https://github.com/giuseppemasi99/DiffCATS

Iller River at Kempten, the Danube River at Dillingen, and the Isar River at Lenggries between the year 2017 and 2019. The Iller is a tributary of the Danube, and we expect that an increase in the water level of the former will flow into the latter within a day, i.e., with a lag of 1 time-step. In this case, $d = 3$ and the only causal relationship is $x_{t-1}^{\text{Kempten}} \Rightarrow x_t^{\text{Dillingen}}$.

• **Air Quality Index** (AQI): introduced by Cheng et al. (2024), it consists of the PM2.5 pollution index monitored hourly ($\delta t = 1$ hour) over the course of one year by 36 stations spread across Chinese cities. In this case, $d = 36$ and the available causal relationships are modeled through a Granger Causality matrix, which is based on the pairwise distances between sensors.

• **Hénon**: introduced by Li et al. (2023), this synthetic dataset consists of $d = 6$ coupled Hénon chaotic maps (Kugiumtzis, 2013) in which there is one positive ($x_{t-2}^i \Rightarrow x_t^i$) and two negative causal relationships ($-x_{t-1}^i \Rightarrow x_t^i$ and $-x_{t-1}^i \Rightarrow x_t^{i+1}$), $\forall\, 1 \leq i \leq d$. The equations generating this dataset are described in Appendix B.1.

In the experiments, the maximum lag $\tau_{max}$, introduced in Section 4.2, is fixed to 2 for all the datasets, consistent with the maximum lag observed in the ground truth. A sensitivity analysis, and a discussion, of parameter $\tau_{max}$ is provided in Appendix C.3. The sequence length is 32 for the Hénon and the Rivers datasets, and 24 for the AQI dataset.

## 5.2   Models

**Benchmarks.**   We compare our model against the two most recent state-of-the-art works described in Section 2, namely CAUSALTIME (Cheng et al., 2024) and CR-VAE (Li et al., 2023). We also include two other baseline solutions using diffusion models: (i) BASE-DIFFUSION, a vanilla diffusion framework applied to time-series generation; (ii) CSDI (Tashiro et al., 2021), appropriately adapted to a generation task as described by Coletta et al. (2023). A detailed description of these models can be found in Appendix B.2.
`DiffCATS`: We trained our model setting the $\lambda$ parameters in Equation (2) and Equation (3) to $\lambda_1 = 0.01$ and $\lambda_2 = 1$, respectively. An extensive ablation study about the impact of each loss component along with additional loss functions, namely the $\ell_1$-norm and a Fourier-based loss, is presented in Appendix C.2. The full list of hyper-parameters is shown in the Appendix in Table 7. To extract the causal graph from the VAR coefficients, we set $\rho = 1\%$ and $q = 0.95$ (see Definition 1).

## 5.3   Evaluation Metrics

**Evaluation of time-series.**   We evaluated the quality of the synthetic time-series using the following metrics for fidelity, usefulness, and diversity: DISCRIMINATIVE SCORE (Discr.) (Yoon et al., 2019), PREDICTIVE SCORE (Pred.) (Yoon et al., 2019), AUTHENTICITY (Auth.) (Alaa et al., 2022), MAXIMUM MEAN DISCREPANCY (MMD) (Gretton et al., 2006) and CROSS-CORRELATION (xCorr.). All of them are well described in Appendix B.4.

**Evaluation of Causal Graphs.**   We note that, despite the existence of a causal phenomenon relating the variables of the datasets, not all the samples extracted from the long time-series may exhibit clear evidence of this. For instance, concerning the Rivers dataset, even if the Iller is a tributary of the Danube, if there is no significant variation in the water level of the former, the phenomenon of causality cannot be observed. Indeed, the water level of the three rivers simply remains stable over substantial periods of time. As a result, many time-step windows extracted from the dataset will not provide evidence of the causal relationship. Considering this issue, we employ metrics that do not penalize missing causal relationships from a sample. In contrast, we want to penalize those causal relationships identified by the model when domain knowledge clearly indicates that such relationships do not exist in real data (for instance, a tributary cannot be caused by the recipient river). The recent works of Ahmad et al. (2022) and Hasan et al. (2023) addressed this problem by introducing metrics based on the false positive rate of causal relationships.

Accordingly, we consider the GRANGER CAUSALITY FALSE POSITIVE RATE (GC-FPR) and the CAUSAL GRAPH FALSE POSITIVE RATE (Graph-FPR), which account for the fraction of edges in the graph known to be incorrect. Notice that, since CR-VAE does not output a causal graph for each sample, we use the F1-SCORE to evaluate its causal relationships with respect to the ground-truth Granger Causality matrix.

We emphasize that the generative method proposed in this paper is not designed to be used as a causal discovery algorithm, so we do not focus on the exact causal relationships expected to be extrapolated from the datasets. Rather than that, we focus on the realism of the causal graphs and their consistency with the corresponding generated synthetic time-series, ensured by the design of the generative pipeline.

Finally, we evaluated the INFERENCE TIME (Inf. time) of the models, i.e. the time to generate a synthetic sample and the corresponding graph.

### 5.4 Results

In this section, we discuss the results of our experiments. All the quantitative scores are shown in Table 1. We report the results of `DiffCATS` with respect to the state-of-the-art approaches discussed in Section 5.2, namely, BASE-DIFFUSION and CSDI for comparing the generated time-series, and CAUSALTIME and CR-VAE for comparing both time-series and causal graphs. We point out that Base Diffusion and CSDI are included as reference points, to show that `DiffCATS` ' time-series quality is not far from high-performing time-series-only generators, while additionally providing causal graphs. All the results report the mean and standard deviation across 10 different seeds.

Table 1: Results of the models on the three datasets, where ↓ indicates *lower is better* and ↑ indicates *higher is better*. For each metric, the best result is highlighted in bold, and the second-best result is underlined (restricted to models able to generate both time-series and causal graphs).

| Dataset | Metric | Models Time-Series Only | | Models Time-Series & Causal-Graphs | | |
|---|---|---|---|---|---|---|
| | | BASE DIFFUSION | CSDI | DiffCATS | CAUSALTIME | CR-VAE |
| Hénon | Discr. ↓ | $0.032 \pm 0.008$ | $0.026 \pm 0.011$ | $\mathbf{0.032 \pm 0.017}$ | $0.311 \pm 0.142$ | $\underline{0.243 \pm 0.113}$ |
| | Pred. ↓ | $0.170 \pm 0.005$ | $0.221 \pm 0.006$ | $\mathbf{0.156 \pm 0.009}$ | $\underline{0.200 \pm 0.011}$ | $0.244 \pm 0.012$ |
| | Auth. ↑ | $0.679 \pm 0.005$ | $0.905 \pm 0.015$ | $\underline{0.693 \pm 0.008}$ | $\mathbf{0.720 \pm 0.031}$ | $0.651 \pm 0.111$ |
| | MMD ↓ | $0.001 \pm 0.000$ | $0.001 \pm 0.000$ | $\mathbf{0.001 \pm 0.000}$ | $\underline{0.002 \pm 0.000}$ | $0.012 \pm 0.009$ |
| | xCorr ↓ | $0.034 \pm 0.005$ | $0.094 \pm 0.005$ | $\mathbf{0.029 \pm 0.005}$ | $\underline{0.060 \pm 0.022}$ | $0.131 \pm 0.031$ |
| | GC-FPR ↓ | — | — | $\mathbf{0.311 \pm 0.001}$ | $\underline{0.482 \pm 0.041}$ | $0.520 \pm 0.070^*$ |
| | Graph-FPR ↓ | — | — | $\mathbf{0.017 \pm 0.000}$ | $\underline{0.231 \pm 0.010}$ | — |
| | Inf. time ↓ | 1231ms | 241ms | $\underline{1548\text{ms}}$ | 8790ms | $\mathbf{194\text{ms}}^*$ |
| Rivers | Discr. ↓ | $0.074 \pm 0.007$ | $0.019 \pm 0.022$ | $\mathbf{0.067 \pm 0.010}$ | $\underline{0.090 \pm 0.050}$ | $0.110 \pm 0.090$ |
| | Pred. ↓ | $0.030 \pm 0.001$ | $0.043 \pm 0.002$ | $\underline{0.033 \pm 0.001}$ | $\mathbf{0.026 \pm 0.001}$ | $0.036 \pm 0.002$ |
| | Auth. ↑ | $0.586 \pm 0.008$ | $1.000 \pm 0.000$ | $\underline{0.630 \pm 0.010}$ | $0.560 \pm 0.030$ | $\mathbf{0.720 \pm 0.020}$ |
| | MMD ↓ | $0.001 \pm 0.000$ | $0.002 \pm 0.000$ | $\mathbf{0.001 \pm 0.000}$ | $0.009 \pm 0.011$ | $0.059 \pm 0.029$ |
| | xCorr ↓ | $0.014 \pm 0.004$ | $0.018 \pm 0.006$ | $0.020 \pm 0.010$ | $\mathbf{0.010 \pm 0.000}$ | $0.120 \pm 0.020$ |
| | GC-FPR ↓ | — | — | $\mathbf{0.220 \pm 0.000}$ | $0.570 \pm 0.010$ | $0.370 \pm 0.140^*$ |
| | Graph-FPR ↓ | — | — | $\mathbf{0.070 \pm 0.000}$ | $\underline{0.220 \pm 0.010}$ | — |
| | Inf. time ↓ | 1252ms | 239ms | $\underline{1492\text{ms}}$ | 4248ms | $\mathbf{148\text{ms}}^*$ |
| AQI | Discr. ↓ | $0.287 \pm 0.010$ | $0.238 \pm 0.068$ | $0.293 \pm 0.050$ | $0.460 \pm 0.020$ | $\mathbf{0.250 \pm 0.040}$ |
| | Pred. ↓ | $0.043 \pm 0.001$ | $0.048 \pm 0.001$ | $\underline{0.047 \pm 0.001}$ | $0.054 \pm 0.001$ | $\mathbf{0.043 \pm 0.001}$ |
| | Auth. ↑ | $0.860 \pm 0.008$ | $0.980 \pm 0.013$ | $\mathbf{0.820 \pm 0.010}$ | $0.770 \pm 0.010$ | $\underline{0.800 \pm 0.100}$ |
| | MMD ↓ | $0.001 \pm 0.000$ | $0.018 \pm 0.001$ | $\mathbf{0.001 \pm 0.000}$ | $0.008 \pm 0.001$ | $\underline{0.017 \pm 0.001}$ |
| | xCorr ↓ | $0.078 \pm 0.005$ | $0.080 \pm 0.003$ | $\underline{0.070 \pm 0.010}$ | $\mathbf{0.030 \pm 0.010}$ | $0.120 \pm 0.010$ |
| | GC-FPR ↓ | — | — | $\mathbf{0.390 \pm 0.000}$ | $\underline{0.490 \pm 0.000}$ | $0.270 \pm 0.000^*$ |
| | Graph-FPR ↓ | — | — | — | — | — |
| | Inf. time ↓ | 1079ms | 242ms | $\underline{1395\text{ms}}$ | 205s | $\mathbf{442\text{ms}}^*$ |

Considering only the models that are able to generate both the time-series and the corresponding causal graph, we can see that regarding the fidelity and the quality of the synthetic time-series, `DiffCATS` outperforms the other approaches in terms of MMD on all three datasets, maintaining a satisfactory degree of AUTHENTICITY. It is also the best model concerning the DISCRIMINATIVE SCORE on two out of three datasets, and in all the other cases, it obtains scores very close to the benchmark. This validates our generated samples with respect to their originality, usefulness, and indistinguishability from real data.

The results comparing `DiffCATS`' time-series with those generated by the two state-of-the-art models capable of producing only time-series demonstrate a comparable level of quality between them. This ensures that the generation of the time-series through the VAR coefficients does not sacrifice the ability of the model to produce high-quality samples.

Regarding the causal graphs, `DiffCATS` achieves both the best GC-FPR and the best Graph-FPR scores in all three datasets, demonstrating its effectiveness in extracting causal relationships from the VAR coefficients. This result is of critical importance given that it ensures the reliability of the graphs as a representation

of the causal relationships exhibited by the time-series. For the AQI dataset, we report only the GC-FPR metric that evaluates the Granger Causality matrix, as no lag information is provided in the ground-truth causal phenomena. We recall that CR-VAE does not output a causal matrix for each sample, but it is learned and fixed in the trained model. Therefore, since the Granger causality matrix is the same for all generated samples, the GC-FPR metric does not fully capture the model's ability in this context. For this reason we reported the F1-score of the learned matrix with respect to the ground-truth Granger causality matrix, highlighting room for improving performance.

Summarizing, we highlight that our model achieves the lowest DISCRIMINATIVE SCORE and MMD along with the best Graph-FPR, ensuring that the synthetic samples exhibit a high level of realness and the causal graphs are reliable. Furthermore, as evidenced in Figure 1, the model is able to generate synthetic traces with strictly associated causal graphs, eliminating the need for stationarity assumptions in causal relationships employed by previous works.

We also evaluated the inference time of the models to obtain a sample made up of the synthetic time-series and the corresponding causal graph. CR-VAE turned out to be the fastest, thanks to its VAE-based architecture. However, great time saving occurs because the causal graph is fixed for each sample since it is extracted from the parameters of the model. Our model achieves an inference time significantly lower than CAUSALTIME, mainly because the post-processing of the feature importance through DeepSHAP is very time-consuming. Instead, in our architecture, the causal graph is generated simultaneously with the time-series, with only a moderate overhead. Moreover, the sampling of diffusion models can be accelerated, using, for example, implicit diffusion models (Song et al., 2021).

Additional experiments and results can be found in the Appendix, including the evaluation of the time-series through dimensionality reduction techniques, namely $t$-SNE and PCA [C.4], kernel density estimation [C.5], the evolution of the evaluation metrics during the training [C.6], the robustness of the graph extraction to noise in the time-series [C.7.1] and a different approach to extract the causal graph from the VAR coefficients using the Dixon's Q Test [C.7.2].

## 6 Downstream Tasks

### 6.1 Benchmark of Causal Discovery Algorithms

Our primary downstream task is to benchmark causal discovery algorithms. Given a generated couple $\langle \hat{\boldsymbol{x}}, \hat{g} \rangle$, we feed the algorithm with the generated time-series $\hat{\boldsymbol{x}}$ and we compare the predicted causal graph against the generated graph $\hat{g}$. Related work presenting this type of benchmark is described in Appendix E.1. In our benchmark, we included the following approaches:
• **Granger-Causality**-based: Granger Causality (GC, (Granger, 1969)); Neural Granger Causality (NGC, (Tank et al., 2021)); economy-SRU (eSRU, (Khanna & Tan, 2019)); Temporal Causal Discovery Framework (TCDF, (Nauta et al., 2019)); CUTS (Cheng et al., 2022); CUTS+ (Cheng et al., 2023);
• **LiNGAM**-based: ICA-LiNGAM (Shimizu et al., 2006); VARLiNGAM (Hyvärinen et al., 2010); DirectLiNGAM (Shimizu et al., 2011);
• **Constraint**-based: PCMCI+ (Runge et al., 2020);
• **Gradient**-based: NTS-NOTEARS (Sun et al., 2023); DYNOTEARS (Pamfil et al., 2020); Rhino (Gong et al., 2023);
• **CCM**-based: Latent Convergent Cross Mapping (LCCM, (De Brouwer et al., 2020));
• **Other**: Neural Graphical Model (NGM, (Bellot et al., 2021)) employing neural ordinary differential equations.

The results are shown in Table 2, evaluated in terms of AUROC and AUPRC (Area Under Precision-Recall Curve). To always have a well-defined ground-truth, for this benchmark, we selected the strongest 15% causal connections for each sample. As additional experiments, we also executed the benchmark using the top 1% approach described in Section 4.3. These results are reported in Appendix E.3.

The comparison between real and synthetic data performance demonstrates encouraging coherence across multiple causal discovery algorithms, suggesting that the synthetic data effectively captures key characteristics

of real-world causal relationships. Several high-performing methods, including PCMCI+, CUTS, and CUTS+ show remarkably consistent performance between real and synthetic datasets, with AUROC differences typically within 0.05, indicating that the synthetic generation process successfully preserves the essential causal structure and complexity. The synthetic data proves particularly effective for methods like NGM and eSRU, which achieve consistently strong AUPRC scores (0.73-0.81) across both real and synthetic versions, demonstrating the synthetic data's ability to maintain predictive relationships. Notably, the AQI dataset shows the strongest real-synthetic correspondence across methods, while the synthetic Rivers dataset successfully replicates the challenging characteristics of its real counterpart. While some methods, such as NTS-NOTEARS and TCDF, exhibit performance variations, these discrepancies appear more attributable to algorithmic sensitivity rather than fundamental limitations in synthetic data quality. Overall, these results indicate that the synthetic data provides a high-quality approximation of real-world causal dynamics, offering a valuable benchmark that captures the essential complexity needed for robust causal discovery evaluation.

Table 2: Results of the Causal Discovery algorithms on real and synthetic data. For real data, the **highest value** is shown in bold and the second-highest value is underlined. For synthetic data, green background indicates comparable performance to real data, while red background indicates a high difference from real data performance.

| Metric | Method | Real Data | | | Synthetic Data | | |
|--------|--------|-----------|-----------|-----------|-----------|-----------|-----------|
| | | Hénon | Rivers | AQI | Hénon | Rivers | AQI |
| **AUROC** | GC | $0.54 \pm 0.09$ | $0.69 \pm 0.24$ | $0.53 \pm 0.04$ | $0.52 \pm 0.03$ | $0.57 \pm 0.07$ | $0.50 \pm 0.00$ |
| | DYNOTEARS | $0.62 \pm 0.11$ | $0.54 \pm 0.23$ | $0.49 \pm 0.03$ | $0.60 \pm 0.04$ | $0.51 \pm 0.03$ | $0.50 \pm 0.00$ |
| | NTS-NOTEARS | $0.48 \pm 0.00$ | $0.50 \pm 0.00$ | $0.50 \pm 0.01$ | $0.57 \pm 0.04$ | $0.69 \pm 0.10$ | $0.50 \pm 0.00$ |
| | PCMCI+ | $0.70 \pm 0.14$ | $0.71 \pm 0.28$ | $0.72 \pm 0.04$ | $0.74 \pm 0.02$ | $0.77 \pm 0.06$ | $0.74 \pm 0.00$ |
| | Rhino | $0.53 \pm 0.01$ | $0.45 \pm 0.03$ | $0.62 \pm 0.02$ | $0.51 \pm 0.01$ | $0.53 \pm 0.06$ | $0.50 \pm 0.00$ |
| | CUTS | $\mathbf{0.80 \pm 0.10}$ | $0.62 \pm 0.04$ | $\underline{0.78 \pm 0.02}$ | $0.75 \pm 0.02$ | $0.76 \pm 0.06$ | $0.74 \pm 0.00$ |
| | CUTS+ | $\underline{0.76 \pm 0.01}$ | $\underline{0.72 \pm 0.01}$ | $\mathbf{0.79 \pm 0.00}$ | $0.75 \pm 0.02$ | $0.75 \pm 0.08$ | $0.74 \pm 0.00$ |
| | Neural-GC | $0.68 \pm 0.01$ | $0.56 \pm 0.02$ | $0.50 \pm 0.00$ | $0.72 \pm 0.01$ | $0.52 \pm 0.05$ | $0.50 \pm 0.01$ |
| | NGM | $0.60 \pm 0.03$ | $0.68 \pm 0.10$ | $0.53 \pm 0.03$ | $0.61 \pm 0.06$ | $0.69 \pm 0.12$ | $0.50 \pm 0.00$ |
| | LCCM | $0.54 \pm 0.02$ | $0.50 \pm 0.00$ | $0.51 \pm 0.00$ | $0.55 \pm 0.00$ | $0.50 \pm 0.00$ | $0.52 \pm 0.00$ |
| | eSRU | $0.50 \pm 0.00$ | $\mathbf{0.73 \pm 0.07}$ | $0.50 \pm 0.00$ | $0.50 \pm 0.00$ | $0.75 \pm 0.11$ | $0.50 \pm 0.00$ |
| | TCDF | $0.49 \pm 0.02$ | $0.50 \pm 0.02$ | $0.55 \pm 0.02$ | $0.52 \pm 0.03$ | $0.50 \pm 0.01$ | $0.50 \pm 0.00$ |
| | ICA-LiNGAM | $0.45 \pm 0.06$ | $0.72 \pm 0.27$ | $0.52 \pm 0.01$ | $0.44 \pm 0.07$ | $0.76 \pm 0.29$ | $0.49 \pm 0.01$ |
| | VARLiNGAM | $0.50 \pm 0.06$ | $0.56 \pm 0.27$ | $0.59 \pm 0.03$ | $0.50 \pm 0.06$ | $0.62 \pm 0.31$ | $0.48 \pm 0.03$ |
| | DirectLiNGAM | $0.51 \pm 0.06$ | $0.72 \pm 0.29$ | $0.52 \pm 0.01$ | $0.50 \pm 0.06$ | $0.68 \pm 0.31$ | $0.48 \pm 0.01$ |
| **AUPRC** | GC | $0.42 \pm 0.07$ | $0.40 \pm 0.29$ | $0.57 \pm 0.02$ | $0.47 \pm 0.13$ | $0.46 \pm 0.10$ | $0.57 \pm 0.09$ |
| | DYNOTEARS | $0.56 \pm 0.02$ | $0.62 \pm 0.05$ | $0.63 \pm 0.01$ | $0.52 \pm 0.08$ | $0.58 \pm 0.03$ | $0.65 \pm 0.00$ |
| | NTS-NOTEARS | $0.17 \pm 0.00$ | $0.17 \pm 0.00$ | $0.36 \pm 0.03$ | $0.45 \pm 0.07$ | $0.54 \pm 0.13$ | $0.42 \pm 0.10$ |
| | PCMCI+ | $0.63 \pm 0.09$ | $0.65 \pm 0.15$ | $0.74 \pm 0.04$ | $0.68 \pm 0.02$ | $0.64 \pm 0.05$ | $0.73 \pm 0.02$ |
| | Rhino | $0.59 \pm 0.04$ | $0.64 \pm 0.02$ | $0.58 \pm 0.03$ | $0.70 \pm 0.07$ | $0.66 \pm 0.07$ | $0.69 \pm 0.04$ |
| | CUTS | $0.43 \pm 0.07$ | $0.65 \pm 0.05$ | $0.64 \pm 0.03$ | $0.68 \pm 0.02$ | $0.64 \pm 0.05$ | $0.73 \pm 0.00$ |
| | CUTS+ | $\underline{0.76 \pm 0.03}$ | $0.55 \pm 0.05$ | $0.78 \pm 0.03$ | $0.68 \pm 0.02$ | $0.62 \pm 0.08$ | $0.73 \pm 0.00$ |
| | Neural-GC | $0.66 \pm 0.01$ | $0.56 \pm 0.06$ | $0.62 \pm 0.04$ | $0.68 \pm 0.01$ | $0.55 \pm 0.08$ | $0.64 \pm 0.05$ |
| | NGM | $0.75 \pm 0.03$ | $0.73 \pm 0.09$ | $\mathbf{0.81 \pm 0.06}$ | $0.77 \pm 0.05$ | $0.73 \pm 0.11$ | $0.80 \pm 0.05$ |
| | LCCM | $0.68 \pm 0.01$ | $\mathbf{0.77 \pm 0.00}$ | $0.58 \pm 0.00$ | $0.67 \pm 0.00$ | $0.78 \pm 0.00$ | $0.57 \pm 0.00$ |
| | eSRU | $\mathbf{0.76 \pm 0.01}$ | $\underline{0.77 \pm 0.09}$ | $\underline{0.77 \pm 0.00}$ | $0.78 \pm 0.02$ | $0.76 \pm 0.10$ | $0.81 \pm 0.00$ |
| | TCDF | $0.37 \pm 0.09$ | $0.51 \pm 0.02$ | $0.35 \pm 0.02$ | $0.65 \pm 0.14$ | $0.57 \pm 0.03$ | $0.64 \pm 0.07$ |
| | ICA-LiNGAM | $0.34 \pm 0.05$ | $0.51 \pm 0.37$ | $0.30 \pm 0.02$ | $0.33 \pm 0.05$ | $0.61 \pm 0.40$ | $0.28 \pm 0.01$ |
| | VARLiNGAM | $0.36 \pm 0.06$ | $0.33 \pm 0.35$ | $0.39 \pm 0.04$ | $0.36 \pm 0.06$ | $0.44 \pm 0.41$ | $0.27 \pm 0.02$ |
| | DirectLiNGAM | $0.37 \pm 0.07$ | $0.53 \pm 0.39$ | $0.30 \pm 0.01$ | $0.37 \pm 0.06$ | $0.52 \pm 0.42$ | $0.27 \pm 0.01$ |

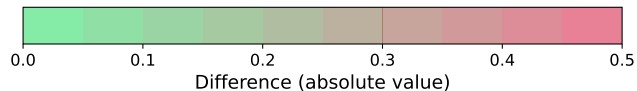

Regarding the performance of the algorithms, three of them — namely, PCMCI+, CUTS, and CUTS+ — achieve the best tradeoff between AUROC and AUPRC on all datasets. Additionally, NGM achieves satisfactory results on the Hénon and Rivers datasets, yielding AUPRC values among the highest. Instead, Neural-GC performed well only on the synthetic dataset of our benchmark, while eSRU performed well only on the Rivers dataset. While LiNGAM-based approaches struggled with the non-linear Hénon dataset,

ICA-LiNGAM and DirectLiNGAM demonstrated competitive performance on the real-world datasets (Rivers and AQI), particularly in terms of AUROC. Among the constraint-based approaches, only PCMCI+ achieved satisfying performances, while, in general, the Granger-Causality-based approaches proved to be the best ones. None of the methods got an AUROC lower than 0.5, meaning that there were no inverted classifications. The overall performance of tested algorithms is lower than what has been reported on simpler synthetic datasets, such as Lorenz-96 (Cheng et al., 2023; Tank et al., 2021), where some methods achieved near-perfect scores. This performance gap may suggest that current algorithms are still imprecise for certain samples and datasets, and they could be further refined to improve accuracy. In general, more challenging synthetic datasets should be used to rigorously test and potentially improve existing TSCD methods. The performance degradation observed in some algorithms when exposed to new data further underscores the need for this approach.

**Dealing with Latent Confounders.** The existence of latent factors, called *confounders*, that influence both the independent variable (the cause) and the dependent variable (the effect) may lead to spurious associations, making it harder to determine the true causal relationship. Our generative model is designed to faithfully replicate the underlying characteristics of the input real dataset, including any statistical and causal properties that are present. As such, if the real dataset exhibits associations induced by latent confounders, synthetic data generated by our framework will reflect these characteristics as well.

Schur & Peters (2024) conducted an experiment in which the observed variable $X$ is modeled as an Ornstein–Uhlenbeck (OU) process, the hidden confounder $U$ as an independent spectrally-sparse OU process, and the effect time-series $Y$ according to the equation $Y_t = \beta X_t + U_t + \eta_t$, where $\eta_t$ is i.i.d. Gaussian noise. The objective is to infer the true causal effect of $X$ on $Y$, i.e. the estimation of $\beta$. The authors show that their DecoR method achieves a lower Mean Absolute Error (MAE) in estimating $\beta$ with respect to a standard Ordinary Least Squares (OLS) approach.

Building on this framework, we assessed the ability of `DiffCATS` to preserve the relevant statistical and causal properties of data generated from the true OU process. Specifically, we trained the generative model on data simulated according to the above OU processes, with the parameters described by Schur & Peters (2024). We evaluate how well causal effect estimation methods could recover $\beta$ on both real (directly simulated) and synthetic (generative model–produced) data. The results in Table 3 indicate that the generative model is able to retain the essential statistical and causal properties of the original OU-based data, as evidenced by the consistent estimation performance of both OLS and DecoR.

Table 3: MAE in estimating $\beta$.

|  | Real | Synthetic |
|---|---|---|
| OLS | $0.76 \pm 0.23$ | $0.78 \pm 0.19$ |
| DecoR | $0.16 \pm 0.14$ | $0.19 \pm 0.15$ |

These findings show that: *(i)* existing deconfounding algorithms can be effectively applied to our synthetic data; *(ii)* synthetic datasets produced by our approach can serve as valuable testbeds for developing and validating new algorithms targeting latent confounding in observational data.

## 6.2 Causal Prediction and Classification

To demonstrate the utility of our dataset beyond benchmarking and improving causal discovery algorithms, we introduce the following two downstream tasks:

- **Causal Prediction**. We show that conditioning a prediction model on the causal graph (see example in Figure 2) significantly improves time-series reconstruction performance, underscoring both the utility and coherence of our graphs. In detail, we trained a 2-layer LSTM to predict the $i$-th feature of a time-series given the remaining features and the corresponding causal graph. Results are shown in Table 4. Additional details are reported in Appendix C.10.

- **Causal Classification**. Classifying observational data (i.e., time-series) based on the underlying system dynamics (i.e., causal graphs) is a critical task across numerous domains. Indeed, causal

classification has broad applicability, ranging from emergency detection (i.e., identify transitions to unsafe system dynamics from time-series data) to customer segmentation (i.e., assign customers to meaningful segments based on their responses to incentives — such as purchase history, engagement, or response to promotions — modeled as causal graphs). We demonstrate that our synthetic dataset can effectively support (augment) the training of classification models, particularly in scenarios where certain classes are underrepresented in real-world data. For demonstration, we organized the synthetic and real Rivers data into 11 distinct classes according to their causal graphs. We then trained a 2-layer LSTM to classify unseen real time-series, achieving an F1-score of 0.69, which highlights the utility of synthetic data. All the details are described in Appendix C.11.

Table 4: Downstream task - Causal Prediction (Mean Absolute Error).

| Dataset | Predictor | |
|---|---|---|
| | w/ Causal Graph | w/o Causal Graph |
| Hénon | $\mathbf{0.13 \pm 0.01}$ | $0.19 \pm 0.01$ |
| Rivers | $\mathbf{0.01 \pm 0.00}$ | $0.02 \pm 0.00$ |

## 7   Conclusions & Limitations

We introduced `DiffCATS`, a novel pipeline to generate faithful time-series along with realistic and coherent causal graphs specifically suited for the TSCD task. To the best of our knowledge, this is the first work to incorporate the simultaneous generation of causal graphs for causally related time-series generation without the need for the stationarity assumption. We demonstrated that our model can effectively generate synthetic datasets to support the causal discovery community in enhancing their algorithms in various domains, learning directly from real-world observational data.

We acknowledge that one limitation of this work is that only linear causal relationships are present in the synthetic samples. In Appendix C.9, we show a possible solution to accommodate non-linear causal links by increasing the polynomial degree employed by the VAR. We also note that `DiffCATS` ' expressivity depends on the chosen maximum lag $\tau_{\max}$, which bounds the longest temporal dependencies the model can represent. Finally, the requirement of generating both the time-series and the causal graph could make its time-series-only metrics trail specialized generators (namely CSDI) and its inference slower due to iterative diffusion sampling with respect to competitors like CR-VAE. So practitioners may prefer CR-VAE for fastest sampling (but with a fixed causal pattern for all synthetic samples), CSDI/time-series-only diffusion models when graphs are unnecessary, and `DiffCATS` when coherent per-sample graphs are needed without expensive post-hoc explanation.

In future works this approach can be extended to add a loss-based guidance of the coefficients so that the generation can be conditioned on a prior-known causal graph. In fact, a key advantage of our approach is the realism and flexibility that diffusion models provide, which allows the implementation of sophisticated conditioning strategies on trained models.

**Acknowledgments**

This paper was prepared for informational purposes in part by the Artificial Intelligence Research group of JPMorgan Chase & Co. and its affiliates ("JP Morgan") and is not a product of the Research Department of JP Morgan. JP Morgan makes no representation and warranty whatsoever and disclaims all liability, for the completeness, accuracy or reliability of the information contained herein. This document is not intended as investment research or investment advice, or a recommendation, offer or solicitation for the purchase or sale of any security, financial instrument, financial product or service, or to be used in any way for evaluating the merits of participating in any transaction, and shall not constitute a solicitation under any jurisdiction or to any person, if such solicitation under such jurisdiction or to such person would be unlawful.

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

# A    Theory

## A.1    Assumptions

Our work makes the following assumptions, aligned with several TSCD algorithms:

- **Markovian Condition:** The joint distribution of the multi-variate time-series can be factorized into $P(\boldsymbol{x}) = \prod_i P(x_i|\mathcal{P}(x_i))$, i.e., every variable is dependent only on its parents.

- **Causal Faithfulness:** It assumes that the relationships between variables in the data faithfully reflect the true causal connections between them.

On the other hand, thanks to the fact that our approach generates the time-series and its strictly associated causal graph, we do not need to assume *causal stationarity*:

- **Causal Stationarity:** It states that all the causal links do not change over time.

## A.2    Relation of Definition 1 to Granger Causality and Transfer Entropy

In this section we provide additional details and intuitions to clarify how our Definition 1 relates to established notions such as Granger causality (GC) and Transfer Entropy (TE).

The intuition behind this is as follows:

1. **Connection to Granger causality.** Our method parameterizes inter-variable temporal dependencies using a VAR-type structure, which is the same modeling family typically used to define linear GC. When trained with an $L_2$ (MSE) objective, the resulting estimation is closely related to least-squares fitting of a linear autoregressive model. Under standard preprocessing/regularity conditions (e.g., standardized variables and limited predictor collinearity), larger VAR coefficients tend to correspond to predictors with larger marginal contributions in reducing prediction error.

2. **GC–TE equivalence under Gaussianity.** We rely on the result of Barnett et al. (2009), which shows that for jointly Gaussian processes (equivalently, linear models with Gaussian residuals), GC and TE quantify the same directed dependence up to a constant factor.

**Connection to Granger causality.**    Our formulation identifies candidate links by thresholding the magnitude of learned VAR coefficients. Let $c_\tau^{i,j}(l)$ denote the coefficient multiplying $x_i(l-\tau)$ in the predictor for $x_j(l)$ at lag $\tau$ and time index $l$ (with $\tau \in \{1, \ldots, \tau_{\max}\}$). In our framework, these coefficients arise from the causal reconstruction module, while the overall model is trained to minimize an $L_2$ reconstruction loss: $\mathcal{L}_{\text{Rec}} = \|x_0 - \hat{x}_0\|_2^2$.

Minimizing MSE encourages the model to fit the conditional mean of each target variable given its predictors, which is consistent with least-squares estimation of a linear VAR. Moreover, under a linear-Gaussian noise interpretation (approximately Gaussian, homoscedastic residuals), minimizing squared error is equivalent to maximizing a conditional Gaussian likelihood.

In linear VAR formulations, GC is defined via the improvement in the prediction of a target when including the past of a candidate driver, relative to a restricted model that excludes it. Concretely, consider predicting the target coordinate $x_j(l)$ from the past window $\{x(l-1), \ldots, x(l-\tau_{\max})\}$. Let the *full* linear predictor include all variables' histories up to $\tau_{\max}$, and let the *restricted* predictor remove the regressor corresponding to $x_i(l-\tau)$ (or, in a stronger form, remove all lags of $x_i$). The "no directed influence at lag $\tau$" null can be expressed as $H_0: c_\tau^{i,j} = 0$, (and the "no Granger causality from $i$ to $j$ up to $\tau_{\max}$" null as $H_0: c_\tau^{i,j} = 0 \ \forall \tau \in \{1, \ldots, \tau_{\max}\}$).

In our *sample-specific, time-varying* setting, the natural analogue is local in time: $H_0(l): c_\tau^{i,j}(l) = 0$. Moreover, since Definition 1 aggregates coefficient magnitudes over time using a quantile operator, an "absence of link" consistent with our extraction rule can be stated as: $H_0: Q_q\big(\{|c_\tau^{i,j}(l)|: \tau_{\max} \le l \le L\}\big) \le \varepsilon$, for a small $\varepsilon$ and $Q_q(\cdot)$ denoting the $q$-quantile.

A common definition of linear GC strength is: $\mathcal{F}_{i \to j} = \log \frac{\text{Var}(\varepsilon_j^{\text{res}})}{\text{Var}(\varepsilon_j^{\text{full}})}$, where $\varepsilon_j^{\text{full}}$ and $\varepsilon_j^{\text{res}}$ denote the residuals of the optimal full vs. restricted linear predictors (with analogous multivariate generalizations using covariance

determinants). Our model is trained to minimize the reconstruction loss (Section 4.1), and the causal reconstruction produces $\hat{x}$ through the VAR coefficients. Minimizing squared reconstruction error is precisely the criterion that (in a linear regression interpretation) minimizes residual variance. Thus, in the linear predictive regime, the learned coefficients $\{c_\tau^{i,j}(l)\}$ parameterize the same type of conditional-mean predictor that underlies Granger-style predictability.

We recognize that, our Definition 1 is *not* an explicit computation of $\mathcal{F}_{i\to j}$, nor a formal hypothesis test. Instead, it is a practical *edge-selection* rule based on the magnitude and persistence of the learned coefficients. Summarizing, this is consistent with GC in the following sense:

- If $c_\tau^{i,j}(l) \equiv 0$ for $\tau_{\max} \leq l \leq L$, then $Q_q(|c_\tau^{i,j}(l)|) = 0$ and the corresponding link will not be selected, matching the Granger-style "no predictive contribution" null.

- Under standardized variables and weak predictor collinearity, $|c_\tau^{i,j}(l)|$ serves as a practical proxy for the predictive importance of regressor $x_i(l-\tau)$ in the conditional-mean model for $x_j(l)$: larger $|c_\tau^{i,j}(l)|$ is typically associated with a larger increase in residual variance when that regressor is removed, and hence with larger Granger-style predictability gains. We explicitly state this as an *interpretability proxy*.

**Equivalence of Granger causality and Transfer Entropy (TE).** Standard GC measures how much the prediction-error variance decreases when a variable (or its past) is added to the predictor set. Transfer entropy (TE) can be written as a conditional mutual information:

$$\mathcal{T}_{i\to j} = I\big(X_i^{\mathrm{past}}; X_j(l) \mid X_{-i}^{\mathrm{past}}\big),$$

where $X_{-i}^{\mathrm{past}}$ denotes the past of all variables except $i$ (up to $\tau_{\max}$).

Barnett et al. (2009) show that for jointly Gaussian processes (equivalently, linear models with Gaussian residuals), linear GC $\mathcal{F}_{i\to j}$ and TE $\mathcal{T}_{i\to j}$ quantify the same directed dependence up to a constant factor. Using natural logarithms, this relationship can be written as $\mathcal{F}_{i\to j} = 2\,\mathcal{T}_{i\to j}$, (with the constant changing if a different logarithm base is used). Therefore, within the linear-Gaussian approximation, stronger directed predictability gains (GC) correspond to stronger information flow (TE). Consistent with the discussion above, our coefficient-thresholding rule should be interpreted as a practical proxy for such directed dependence in the linear-Gaussian regime, rather than an explicit computation of GC/TE in full generality.

## B  Implementation Details

### B.1  Datasets

Table 5 reports the most important statistics of our datasets and Figure 4 shows the real causal graphs of the datasets. We also include additional details not discussed in Section 5.1.

Table 5: Statistics of the datasets.

| Dataset | Number of Training Samples | Sequence Length | Number of Variables | Number of Causal Relations |
|---|---|---|---|---|
| Hénon | 11295 | 32 | 6 | 11 |
| Rivers | 9969 | 32 | 3 | 1 |
| AQI | 7246 | 24 | 36 | 354 |

**Hénon**: As described by Li et al. (2023), it is generated by the following equations:

$$\mathbf{x}_{t+1}^1 = 1.4 - (\mathbf{x}_t^1)^2 + 0.3 \cdot \mathbf{x}_{t-1}^1$$

$$\mathbf{x}_{t+1}^i = 1.4 - (e \cdot \mathbf{x}_t^{i-1} + (1-e) \cdot \mathbf{x}_t^i)^2 + 0.3 \cdot \mathbf{x}_{t-1}^i$$

with $i = 2, \ldots, d$, where the number of dimensions $d = 6$ and $e = 0.3$. The initial values are sampled from a standard Gaussian distribution. In this dataset, the causal graph consists of one positive relationship with a

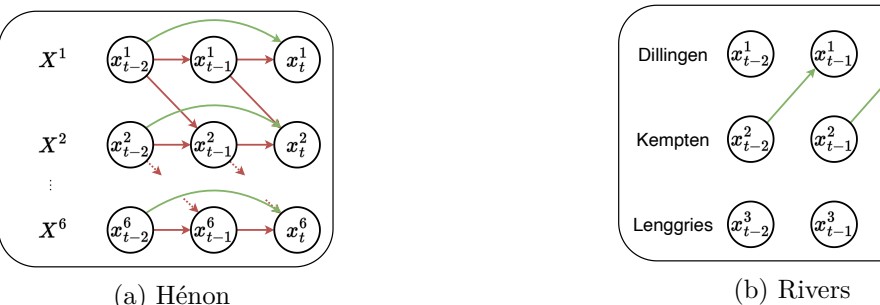

(a) Hénon            (b) Rivers

Figure 4: Real Causal Graphs (green and red arrows represent positive and negative causal relationships, respectively).

lag of 2 between a variable and itself ($x^i_{t-2} \Rightarrow x^i_t$) and two negative relationships. The first one is between the variable and itself with a lag of 1 ($-x^i_{t-1} \Rightarrow x^i_t$); the second one is between two consecutive variables again with a lag of 1 ($-x^i_{t-1} \Rightarrow x^{i+1}_t$).

**Rivers**: The data are provided by the Bavarian Environmental Agency: `https://www.gkd.bayern.de`.

**Air Quality Index** (AQI): It can be found at: `https://www.microsoft.com/en-us/research/project/urban-computing`. We recall that, as in (Cheng et al., 2024) state, the causal relations in the AQI dataset are highly dependent on geometry distances. The graph contained in the dataset they released has been extracted considering the Gaussian kernel and a threshold with respect to the geographic distances of the sensors. In particular,

$$w_{ij} = \begin{cases} 1, & \texttt{dist}(i,j) \leq \sigma \\ 0, & \text{otherwise} \end{cases}$$

where `dist` measures the distance between two sensors and $\sigma$ is set to $\approx 40$ km. See the work of Cheng et al. (2024) for more details.

## B.2 Benchmark

We compare our approach against two state-of-the-art approaches, whose code is available from the respective repositories:

- CAUSALTIME (Cheng et al., 2024): `https://github.com/jarrycyx/UNN`.
  It is an autoregressive model based on normalizing flows, able to observe some time-steps of the time-series and generate the subsequent step. Thanks to this architecture, the authors can extract the importance of each feature in the input time-series using an explainability technique,i.e., DeepSHAP, provided by Sundararajan & Najmi (2020), and eventually extract a causal graph.

- CR-VAE (Li et al., 2023): `https://github.com/hongmingli1995/CR-VAE`.
  It is based on a recurrent VAE made up of a multi-head decoder, in which the $p$-th head is responsible for generating the $p$-th feature of the time-series. Encouraged by a sparsity penalty on the weights of the decoder, it learns a sparse causal matrix able to encode causal relationships among the variables. Since the causal matrix is part of the model's parameter, it will be the same for each synthetic sample generated by the model, in contrast to our approach and CAUSALTIME. Moreover, a notable limitation of CR-VAE is that it is restricted to the notion of Granger Causality, implying that it does not consider the concept of time lag in observing the causal relationships.

- CSDI (Tashiro et al., 2021): `https://github.com/ermongroup/CSDI`. We modified the original CSDI model to make it suitable for the unconditional time-series generation task, following (Coletta et al., 2023).

We tuned the hyper-parameters of these models on all the datasets and they are reported in Table 6.

Table 6: Hyper-parameters of CAUSALTIME, CR-VAE and CSDI.

| Model | Hyper-parameter | Dataset | | |
|---|---|---|---|---|
| | | Hénon | Rivers | AQI |
| CAUSALTIME | Share type | Decoder | Decoder | Decoder |
| | N. Epochs Train Phase 1 | 40 | 20 | 10 |
| | N. Epochs Train Phase 2 | 10 | 10 | 5 |
| | Learning rate | 0.001 | 0.0001 | 0.0001 |
| | Batch size | 32 | 32 | 32 |
| | Hidden_size | 128 | 128 | 128 |
| | N. Layers | 2 | 2 | 2 |
| | N. Heads | 4 | 4 | 4 |
| | Dropout p | 0.1 | 0.1 | 0.1 |
| | Flow length | 4 | 4 | 4 |
| CR-VAE | Hidden | 64 | 64 | 64 |
| | N. Iterations Train Phase 1 | 1000 | 1000 | 1000 |
| | N. Iterations Train Phase 2 | 90000 | 9000 | 90000 |
| | Learning rate | 0.05 | 0.05 | 0.05 |
| | Batch size | 1024 | 1024 | 1024 |
| CSDI | Epochs | 39 | 79 | 59 |
| | Batch size | 16 | 16 | 16 |
| | Learning rate | 0.0001 | 0.0001 | 0.0001 |
| | $\beta$ Schedule | Quadratic | Quadratic | Quadratic |
| | $\beta$ Start | $1e-06$ | $1e-06$ | $1e-06$ |
| | $\beta$ End | 0.5 | 0.5 | 0.5 |
| | Diffusion Timesteps (T) | 50 | 50 | 50 |

## B.3 Architecture of Denoising Network

Here we discuss the denoising network architecture. We slightly modified the architecture of the work of Song et al. (2021) released in the authors' repository[6] by adding the convolution layer to output the coefficients. The overall architecture of $\text{DEN}_\theta$ is depicted in Figure 5. It consists of an initial convolution layer and a series of RESNET and ATTENTION blocks shown in Figures 6 and 7, respectively.

To represent the denoising time-step $t$ the model employs cosine embedding and an MLP block made up of 2 linear layers with the activation function $f(x) = x \cdot \sigma(x)$ in the middle, where $\sigma$ represents the SIGMOID function $\sigma(x) = \frac{1}{1+e^{-x}}$. The time-step information is injected in all the RESNET blocks.

In the pictures, NON-LINEARITY and GROUPNORM refer to the function $f(x)$ and Group Normalization, respectively. The DOWNSAMPLE block is just a $1d$-convolution with a stride equal to 2. The UPSAMPLE block is made up of a Nearest Interpolation and a $1d$-convolution.

The end part of the architecture is made up of $d + 1$ convolutional layers. The first one is responsible for outputting the first $\tau_{max}$ steps of the time-series. Each of the remaining $d$ convolutional layers is responsible for the coefficients related to the causality impact on one feature. Then, the coefficients are multiplied with the initial steps of the time-series and the final output is reconstructed following the formalization in Section 4.2.

The most important hyper-parameters are reported in Table 7.

Table 7: Hyper-parameters of the generative model.

| Training epochs | Batch size | Learning rate | Diffusion Timesteps (T) | $\beta$ schedule | Time-step embedding |
|---|---|---|---|---|---|
| 50 | 32 | 1e-4 | 100 | Linear start=0.0001, end=0.02 | Cosine dim=128 |

## B.4 Evaluation Metrics

In this section, we describe each evaluation metric in detail.

• DISCRIMINATIVE SCORE (Yoon et al., 2019): It measures the fidelity of synthetic time-series, evaluating to which extent they are indistinguishable from real ones. It consists in training an off-the-shelf 2-layer LSTM

---

[6]https://github.com/mirthAI/Fast-DDPM

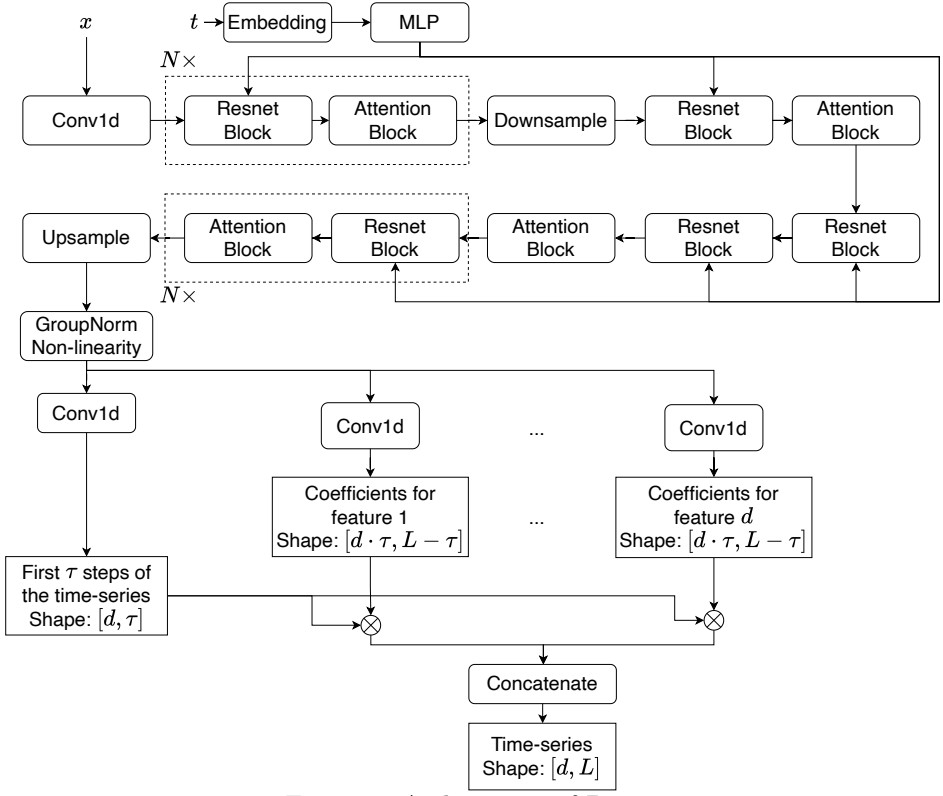

Figure 5: Architecture of $\textsc{Den}_\theta$.

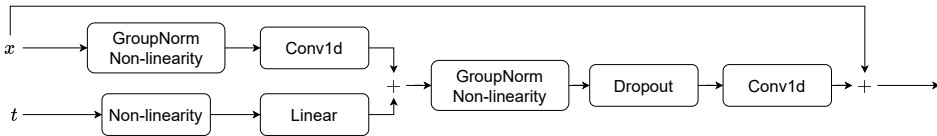

Figure 6: Architecture of the Resnet Block of $\textsc{Den}_\theta$.

to distinguish real samples from synthetic ones. The model is trained for 30 epochs with a learning rate of $1e-4$, hidden size equals 8, and batch size set to 32. The loss function to be optimized is the Binary Cross Entropy where real samples are labeled as 1 and synthetic samples as 0. The score is formally defined as $|0.5 - \text{AUROC}|$, where AUROC is the area under the ROC (Receiver-Operating Characteristic) curve of the trained discriminator.

• Predictive Score (Yoon et al., 2019): It measures the usefulness of synthetic time-series for a downstream prediction task. Following the *train-on-synthetic* and *test-on-real* criterion, a post-hoc sequence-prediction model is trained to predict the subsequent steps of a time-series on synthetic data and evaluated on real data in terms of the Mean Absolute Error (MAE) of the reconstructions. We trained a 2-layer LSTM-based predictor to forecast the last $\frac{1}{10} \cdot \texttt{seq\_len}$ time-steps over each synthetic sample for 10 epochs, with a learning rate of $1e-3$, hidden size equal to 32, and batch size set to 32. The loss function to be optimized is the $\ell_1$-loss. Then, the predictor is evaluated on real data, and the error is quantified using the Mean Absolute Error (MAE). Formally, given a real sequence $\boldsymbol{x}$ of length $\texttt{seq\_len}$ let $\boldsymbol{x}_{first}$ and $\boldsymbol{x}_{last}$ be the first $\frac{9}{10} \cdot \texttt{seq\_len}$ and the last $\frac{1}{10} \cdot \texttt{seq\_len}$ time-steps, respectively. The predictor observe $\boldsymbol{x}_{first}$ and

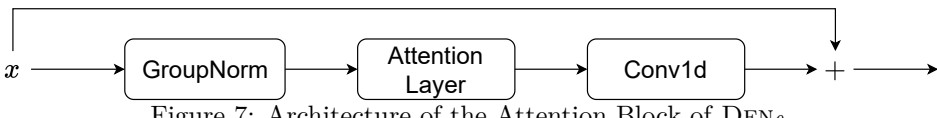

Figure 7: Architecture of the Attention Block of $\textsc{Den}_\theta$.

predicts the subsequent $\frac{1}{10} \cdot$ `seq_len` time-steps, denoted as $\tilde{\boldsymbol{x}}_{pred}$. The MAE-based performance consists of $\frac{1}{\frac{1}{10} \cdot \text{seq\_len}} \sum_{t=1}^{\frac{1}{10} \cdot \text{seq\_len}} |\boldsymbol{x}_{last}(t) - \tilde{\boldsymbol{x}}_{pred}(t)|$.

• AUTHENTICITY (Alaa et al., 2022): It measures the proportion of synthetic data $A \in [0, 1]$ that is *authentic*, i.e. the models should not simply memorize the training dataset by generating copies of real samples it has observed, should instead *create* new, original samples. We considered the original implementation provided by the work of Alaa et al. (2022). The metric is evaluated through a hypothesis test for data copying, which employs a nearest-neighbor classifier. A synthetic sample is considered unauthentic if it is closest to a real training sample. A score close to 1 indicates that the model is generating novel, unseen data.

• MAXIMUM MEAN DISCREPANCY (Gretton et al., 2006): It measures the similarity of synthetic and real time-series distributions. Formally, it is defined as $\text{MMD}^2(P, Q) = \mathbb{E}_P[k(X, X)] - 2 \cdot \mathbb{E}_{P,Q}[k(X, Y)] + \mathbb{E}_Q[k(Y, Y)]$ where $k(\cdot, \cdot)$ is the Radial Basis Function (RBF) kernel. We used the scikit-learn[7] implementation of the RBF kernel.

• CROSS-CORRELATION: It measures the extent to which multiple synthetic time-series preserve the cross-correlation present in real data. Specifically, we evaluate the MAE between the correlation values of the real and synthetic features. We computed the Cross-Correlation distance for each lag up to 4. Formally, let $\boldsymbol{x}$ and $\hat{\boldsymbol{x}}$ be a real and a synthetic sample respectively. Moreover, let $\boldsymbol{x}_i$ and $\hat{\boldsymbol{x}}_i$ be the $i$-th feature of the real and the synthetic sample ($\forall 1 \leq i \leq d$), respectively. The score is formally defined as $\sum_{\tau=0}^{4} \frac{1}{\binom{d}{2}} \cdot \sum_{\{i,j\} \in \binom{\{1,\dots,d\}}{2}} |(\boldsymbol{x}_i \star \boldsymbol{x}_j)(\tau) - (\hat{\boldsymbol{x}}_i \star \hat{\boldsymbol{x}}_j)(\tau)|$, where $(\boldsymbol{x}_i \star \boldsymbol{x}_j)(\tau)$ denotes the cross-correlation between $\boldsymbol{x}_i$ and $\boldsymbol{x}_j$ with respect to lag $\tau$.

## C  Additional Results

### C.1  Samples

Figure 8 visually shows the algorithm for extracting the causal graph from the coefficients samples of the Rivers dataset, according to Definition 1. In the figure, (a) and (b) are two synthetic samples with different causal dynamics. Figure (c) shows the coefficients generating the Dillingen feature. In particular, for both samples, each histogram considers the impact of each of the other features on Dillingen. For instance, the pink histograms represent the impact of Kempten with a lag of 1 time-step on Dillingen since they are the coefficients of the 1 time-step lagged Kempten that are used to generate Dillingen. Similarly to Figure (c), Figures (d) and (e) show the coefficients weighting the impact on the Kempten and the Lenggries features. The dashed black line in the figures is the $q$-quantile ($q = 0.95$)used in Definition 1, employed to quantitatively evaluate the generated coefficients over the time-series time-steps. All the $q$-quantiles are then reported in (f) and (g), for Sample A and Sample B, respectively, where the highest (in terms of magnitude) coefficients are selected to get the $\rho\%$ sparsity parameter used in Definition 1. Indeed, the dashed black line in Figures (f) and (g) is the threshold such that only the $\rho\%$ highest values are selected over the entire set of generated samples.

Figures 9 and 10 show additional examples of real and generated samples for the Rivers and Hénon datasets, respectively.

### C.2  Additional ablation studies

Table 8 show the quantitative results for the other variants of our model, highlighting the impact of each loss component. Additionally, `DiffCATS` W/L1 W/DTW considers a DTW-based loss and a $\ell_1$-norm to regularize the coefficients; `DiffCATS` W/L2 W/FOURIER considers a Fourier-based loss (weighted by a term $\lambda_3 = 100$), and $\ell_2$-norm to regularize the coefficients. We considered the Fourier-based term employed by Yuan & Qiao (2024):

$$\mathcal{L}_{Fourier}(\boldsymbol{x}_0, \hat{\boldsymbol{x}}_0; \theta) = \|\mathcal{FFT}(\boldsymbol{x}_0) - \mathcal{FFT}(\hat{\boldsymbol{x}}_0)\|_2^2, \tag{4}$$

where $\mathcal{FFT}(\cdot)$ indicates the Fast Fourier Transformation (Elliott & Rao, 1982).

---

[7]https://scikit-learn.org/

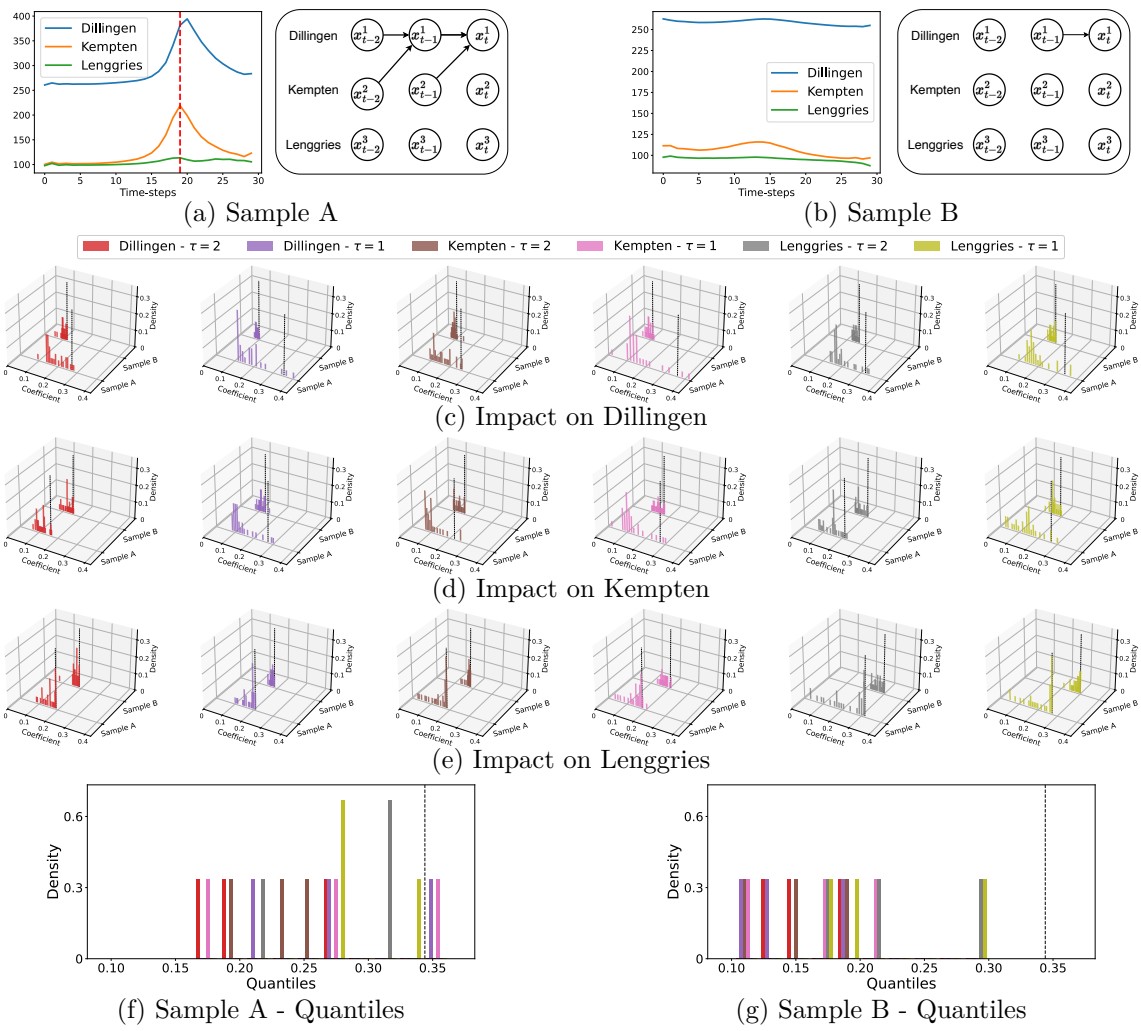

Figure 8: Procedure to extract the causal graphs from the coefficients.

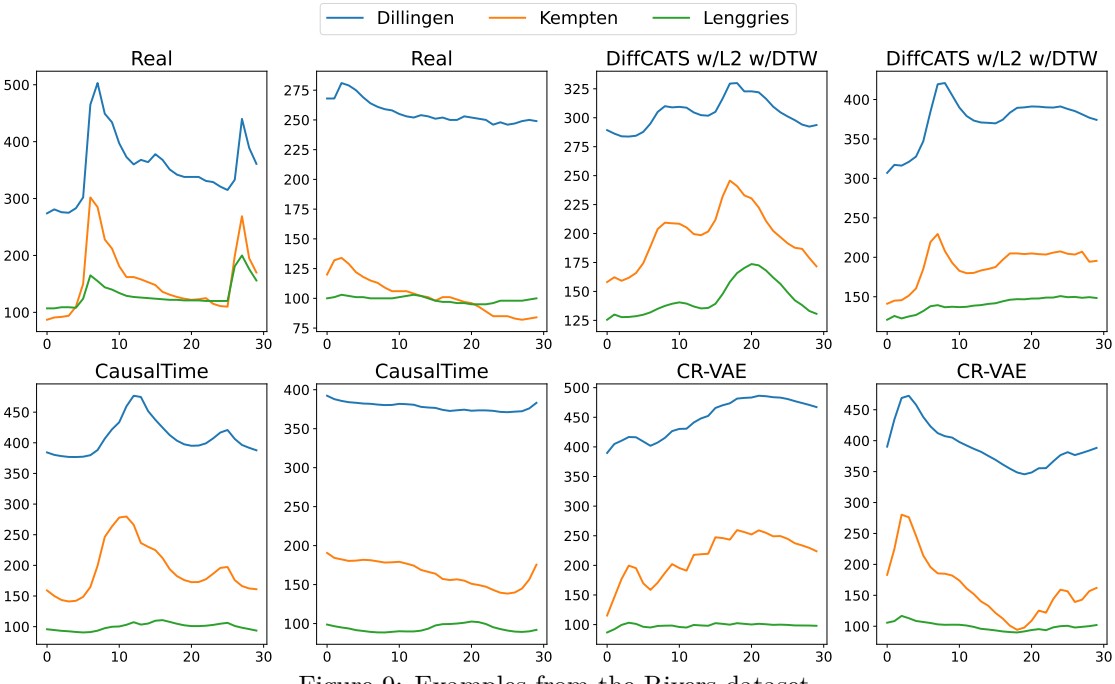

Figure 9: Examples from the Rivers dataset.

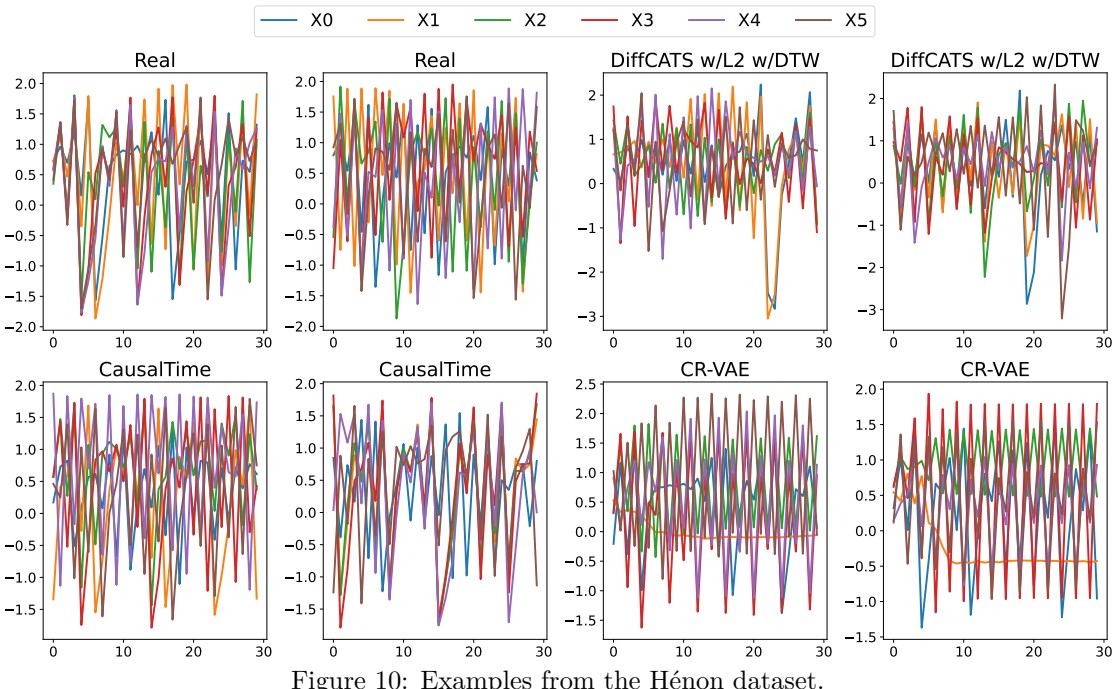

Figure 10: Examples from the Hénon dataset.

Table 8: Results of other models on the three datasets. ↓ indicates *lower is better* and ↑ indicates *higher is better*.

| Dataset | Metric | Models Time-Series & Causal-Graphs | | | | |
|---|---|---|---|---|---|---|
| | | `DiffCATS` | `DiffCATS` w/L2 | `DiffCATS` w/DTW | `DiffCATS` w/L1 w/DTW | `DiffCATS` w/L2 w/Fourier |
| Hénon | Discr. ↓ | $0.041 \pm 0.026$ | $0.045 \pm 0.025$ | $0.038 \pm 0.022$ | $0.035 \pm 0.015$ | $0.040 \pm 0.024$ |
| | Pred. ↓ | $0.216 \pm 0.010$ | $0.217 \pm 0.008$ | $0.216 \pm 0.010$ | $0.217 \pm 0.008$ | $0.217 \pm 0.0010$ |
| | Auth. ↑ | $0.663 \pm 0.006$ | $0.713 \pm 0.007$ | $0.662 \pm 0.007$ | $0.677 \pm 0.008$ | $0.700 \pm 0.006$ |
| | MMD ↓ | $0.001 \pm 0.000$ | $0.001 \pm 0.000$ | $0.001 \pm 0.000$ | $0.001 \pm 0.000$ | $0.001 \pm 0.000$ |
| | xCorr. ↓ | $0.031 \pm 0.003$ | $0.035 \pm 0.004$ | $0.029 \pm 0.004$ | $0.037 \pm 0.005$ | $0.035 \pm 0.05$ |
| | GC-FPR. ↓ | $0.403 \pm 0.001$ | $0.421 \pm 0.001$ | $0.423 \pm 0.001$ | $0.397 \pm 0.000$ | $0.403 \pm 0.001$ |
| | Graph-FPR. ↓ | $0.090 \pm 0.000$ | $0.080 \pm 0.000$ | $0.023 \pm 0.000$ | $0.040 \pm 0.000$ | $0.090 \pm 0.000$ |
| Rivers | Discr. ↓ | $0.080 \pm 0.010$ | $0.130 \pm 0.010$ | $0.070 \pm 0.023$ | $0.480 \pm 0.010$ | $0.100 \pm 0.010$ |
| | Pred. ↓ | $0.035 \pm 0.001$ | $0.037 \pm 0.001$ | $0.041 \pm 0.002$ | $0.043 \pm 0.001$ | $0.036 \pm 0.001$ |
| | Auth. ↑ | $0.580 \pm 0.010$ | $0.620 \pm 0.010$ | $0.585 \pm 0.012$ | $0.870 \pm 0.020$ | $0.610 \pm 0.010$ |
| | MMD ↓ | $0.001 \pm 0.000$ | $0.001 \pm 0.000$ | $0.001 \pm 0.000$ | $0.054 \pm 0.003$ | $0.001 \pm 0.000$ |
| | xCorr. ↓ | $0.060 \pm 0.000$ | $0.060 \pm 0.010$ | $0.022 \pm 0.004$ | $0.080 \pm 0.010$ | $0.070 \pm 0.010$ |
| | GC-FPR. ↓ | $0.230 \pm 0.000$ | $0.220 \pm 0.000$ | $0.158 \pm 0.001$ | $0.270 \pm 0.000$ | $0.230 \pm 0.000$ |
| | Graph-FPR. ↓ | $0.100 \pm 0.000$ | $0.070 \pm 0.000$ | $0.074 \pm 0.002$ | $0.160 \pm 0.000$ | $0.070 \pm 0.000$ |
| AQI | Discr. ↓ | $0.410 \pm 0.020$ | $0.430 \pm 0.010$ | $0.360 \pm 0.015$ | $0.440 \pm 0.010$ | $0.380 \pm 0.030$ |
| | Pred. ↓ | $0.048 \pm 0.001$ | $0.048 \pm 0.001$ | $0.060 \pm 0.002$ | $0.049 \pm 0.001$ | $0.050 \pm 0.001$ |
| | Auth. ↑ | $0.810 \pm 0.020$ | $0.810 \pm 0.020$ | $0.883 \pm 0.008$ | $0.820 \pm 0.010$ | $0.810 \pm 0.010$ |
| | MMD ↓ | $0.001 \pm 0.000$ | $0.001 \pm 0.000$ | $0.001 \pm 0.000$ | $0.001 \pm 0.000$ | $0.001 \pm 0.000$ |
| | xCorr. ↓ | $0.090 \pm 0.010$ | $0.110 \pm 0.010$ | $0.100 \pm 0.003$ | $0.130 \pm 0.010$ | $0.100 \pm 0.010$ |
| | GC-FPR. ↓ | $0.480 \pm 0.000$ | $0.400 \pm 0.000$ | $0.390 \pm 0.000$ | $0.390 \pm 0.000$ | $0.400 \pm 0.000$ |
| | Graph-FPR. ↓ | — | — | — | — | — |

As the results show, incorporating the $\ell_2$-norm of the coefficients into the objective loss as an attempt to sparsify the causal graph reduces the number of wrong connections. Moreover, the DTW-based loss considerably aids in extracting synchronization signals among the temporal sequences, significantly improving overall performance.

## C.3 Sensitivity Analysis of Lag Parameters $\tau_{max}$

In this subsection we perform a sensitivity analysis by varying $\tau_{max}$ on the *rivers* dataset. In general, the parameter $\tau_{max}$ should be selected according to the time resolution and the expected causal dynamics of the underlying phenomena. However, when the true value is uncertain, a slight overestimation can be safely adopted. As shown in Table 9, the model maintains strong performance for both the generated time series and the causal graph, even when $\tau_{max}$ is overestimated or deviates from the true lag of the causal relationships (the maximum ground-truth lag is 2). This demonstrates the model's robustness in generating reliable synthetic data, even with imperfect domain knowledge.

Table 9: Model performance under different lag values.

| $\tau_{max}$ | Discr. score | Pred. score | Graph-FPR |
|---|---|---|---|
| 1 | 0.099 | 0.032 | 0.022 |
| 2 | 0.067 | 0.033 | 0.070 |
| 3 | 0.060 | 0.031 | 0.078 |
| 4 | 0.050 | 0.033 | 0.086 |

## C.4 Dimensionality Reduction

It is used to evaluate the diversity of synthetic samples, i.e., they cover the full variability of real samples. We employed *t*-SNE (Van der Maaten & Hinton, 2008) and PCA (Bryant & Yarnold, 1995) on both real and synthetic data to easily visualize how similar the two distributions are in a 2-dimensional space. We used the scikit-learn[7] implementation for both PCA and *t*-SNE. For each sample, we flattened the dimension of the features by computing the mean.

Figure 11 and Figure 12 show the *t*-SNE and PCA plots of our best model against the state-of-the-art approaches. It can be observed that the distribution of the synthetic samples closely resembles the real one

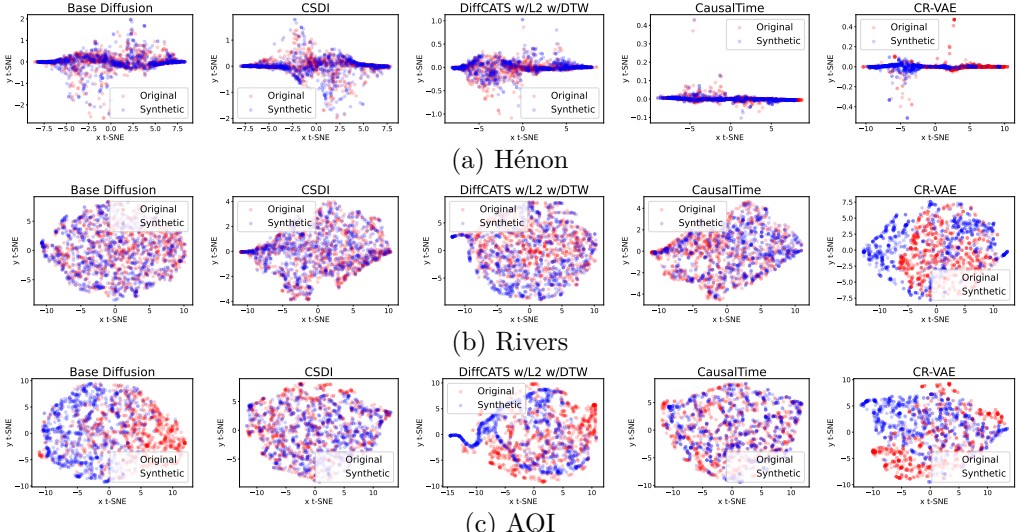

Figure 11: Dimensionality reduction: *t*-SNE.

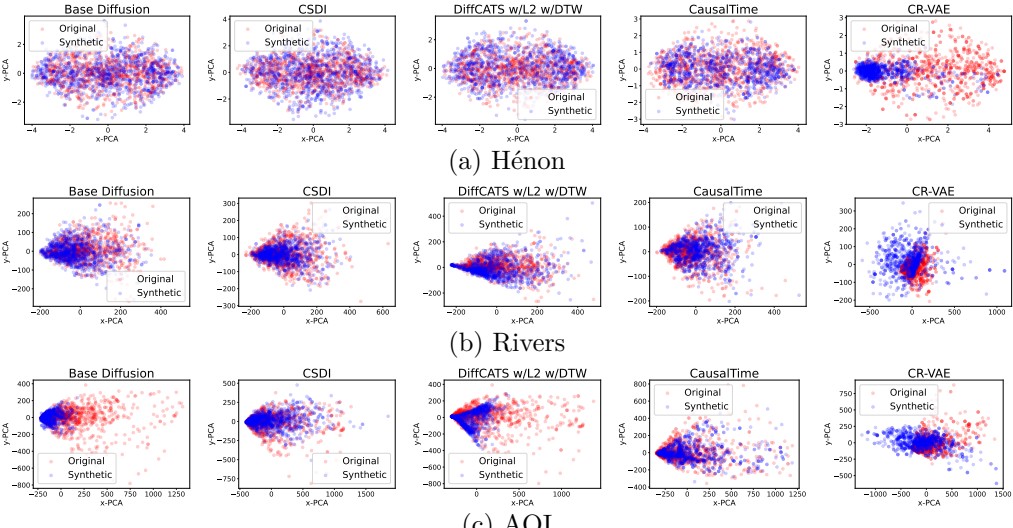

Figure 12: Dimensionality reduction: PCA.

in two of the three datasets (Hénon and Rivers). This visually ensures that the model is generating realistic time-series in a diverse set of fields. Figure 13 and Figure 14 show the dimensionality reduction results for the other variants of the model on all considered datasets.

## C.5 Kernel Density Estimation

Figure 15 shows the data distributions drawn from kernel density estimation (KDE) of **DiffCATS** against the state-of-the-art approaches (ablations in Figure 16). KDE provides a visual assessment of the model's ability to capture the marginal distributions of the real data. A key insight from these plots is the high degree of overlap between the Original real data (red) and **DiffCATS**' data (blue) curves. This indicates that our model faithfully reproduces the statistical properties of the underlying data, including multi-modal distributions and peak densities. In contrast, baseline methods such as CR-VAE often exhibit smoother distributions that fail to capture the sharper density peaks observed in the Rivers and Hénon datasets, suggesting a tendency to over-regularize or average out specific data characteristics. The tight alignment achieved by **DiffCATS**

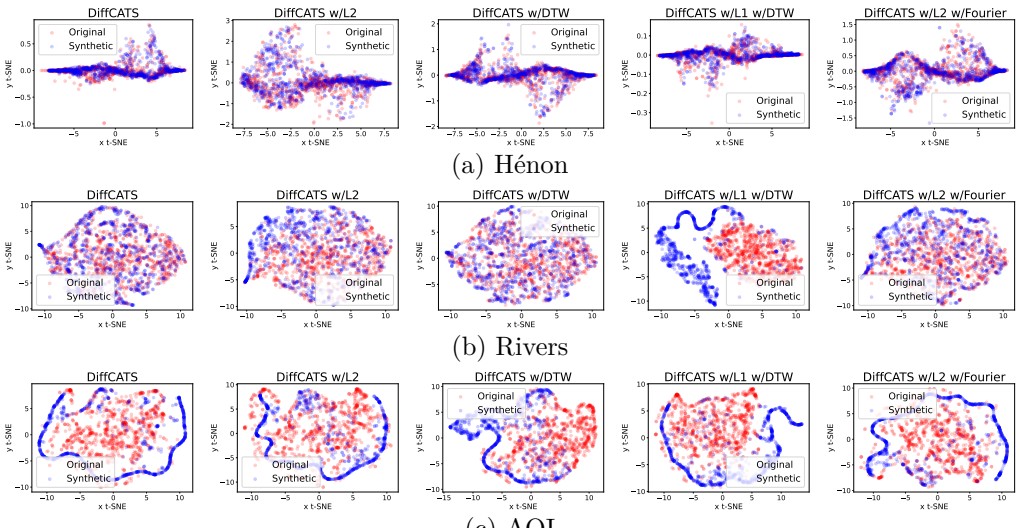

Figure 13: Ablation - Dimensionality reduction: $t$-SNE.

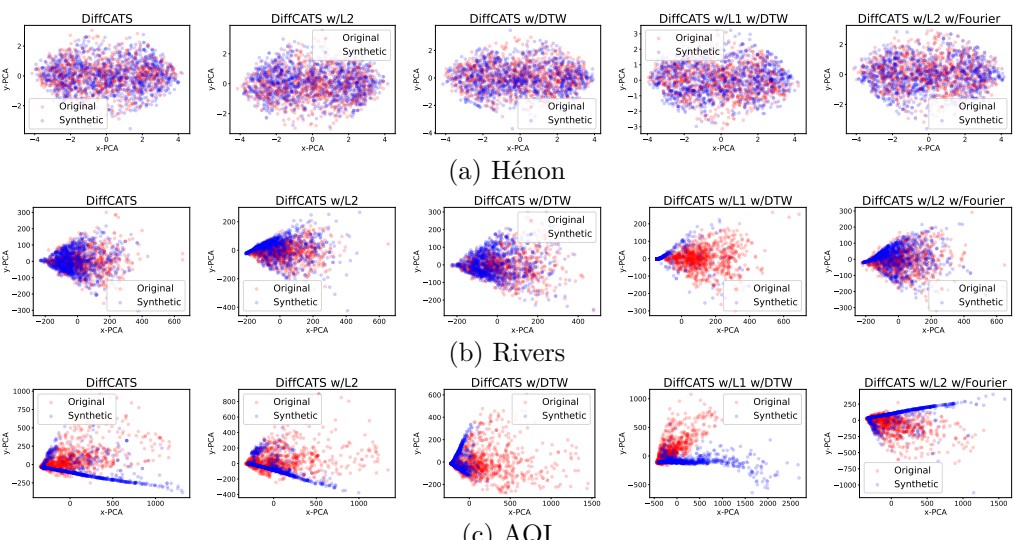

Figure 14: Ablation - Dimensionality reduction: PCA.

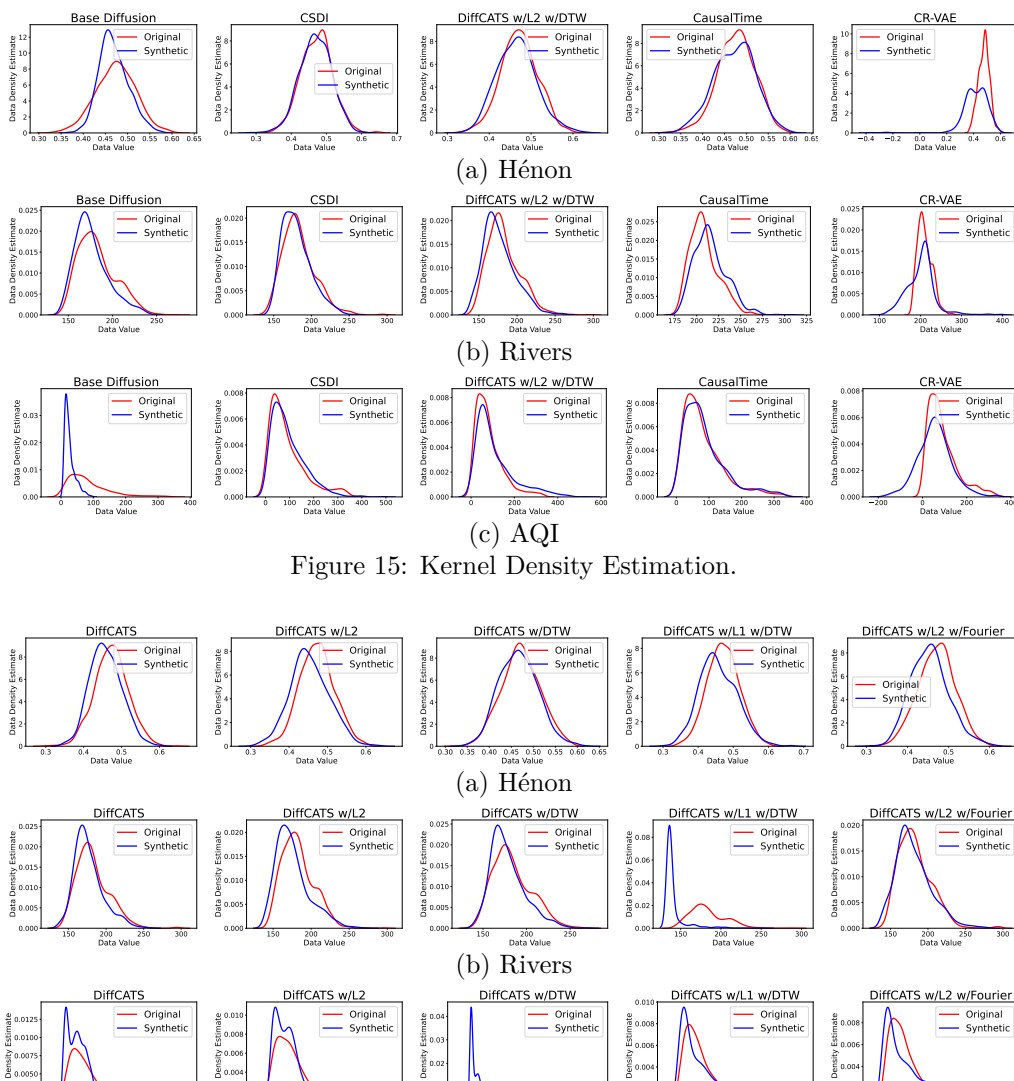

Figure 15: Kernel Density Estimation.

Figure 16: Ablation - Kernel Density Estimation.

confirms that the generated time-series preserve the fundamental distributional nature of the real-world phenomena.

## C.6 Evaluation Metrics during Training

Figures 17 to 19 show the evolution of the evaluation metrics during training on the Hénon, Rivers, and AQI datasets, respectively. This offers several insights into the learning stability and convergence of `DiffCATS`.

First, we observe a rapid convergence of the MMD and Discriminative Score, which drop significantly within the first 20 epochs. This indicates that the diffusion model quickly learns to generate samples that are statistically similar to, and difficult to distinguish from, real data.

Second, the Authenticity metric, while decreasing from an initial value of 1.0 (where noise is purely novel), stabilizes at a robust level (typically between 0.6 and 0.7). This confirms that while the model learns to approximate the real distribution, it does not simply memorize the training set; it continues to generate novel samples rather than copies.

Finally, the causal metrics (GC-FPR and Graph-FPR) show a consistent downward trend or stabilization concurrent with the time-series quality metrics. This demonstrates the effectiveness of the joint optimization strategy: the model successfully refines its understanding of the causal structure (the causal graph) simultaneously with the temporal dynamics, without one objective destabilizing the other.

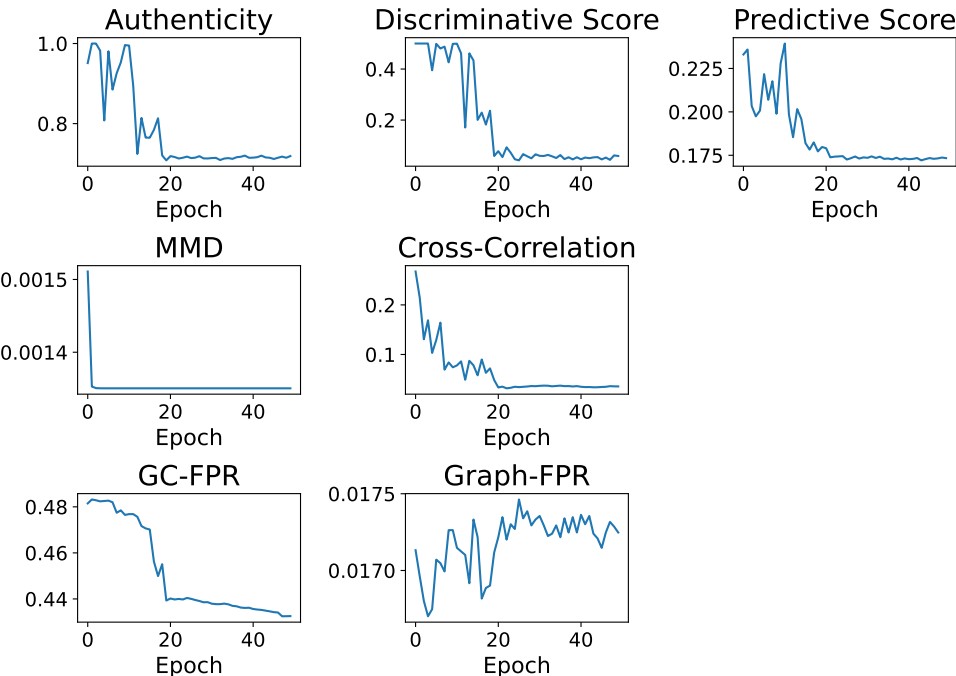

Figure 17: Evaluation metrics during training - Hénon dataset.

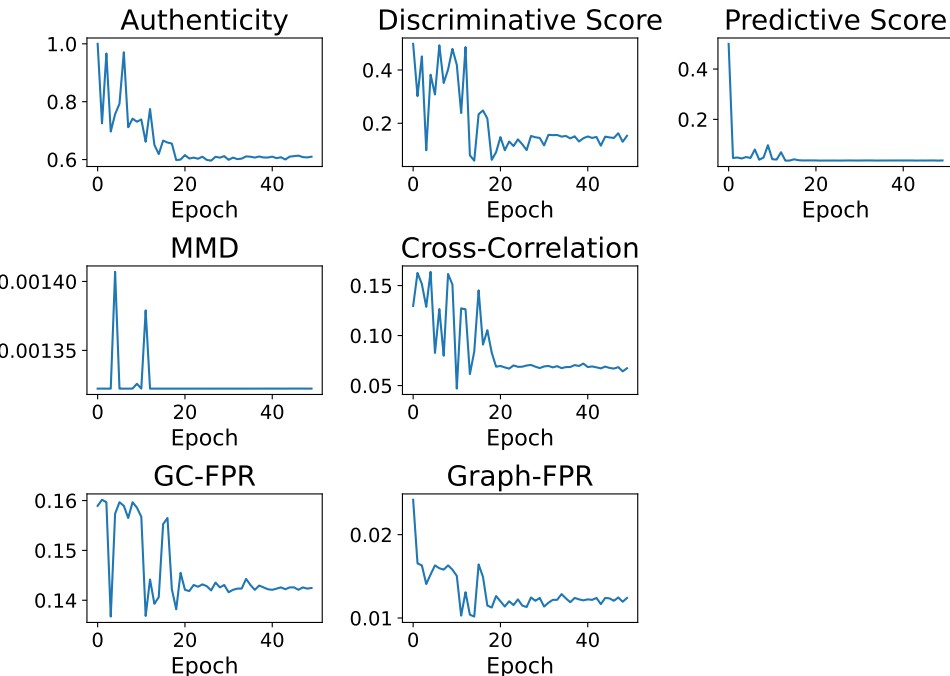

Figure 18: Evaluation metrics during training - Rivers dataset.

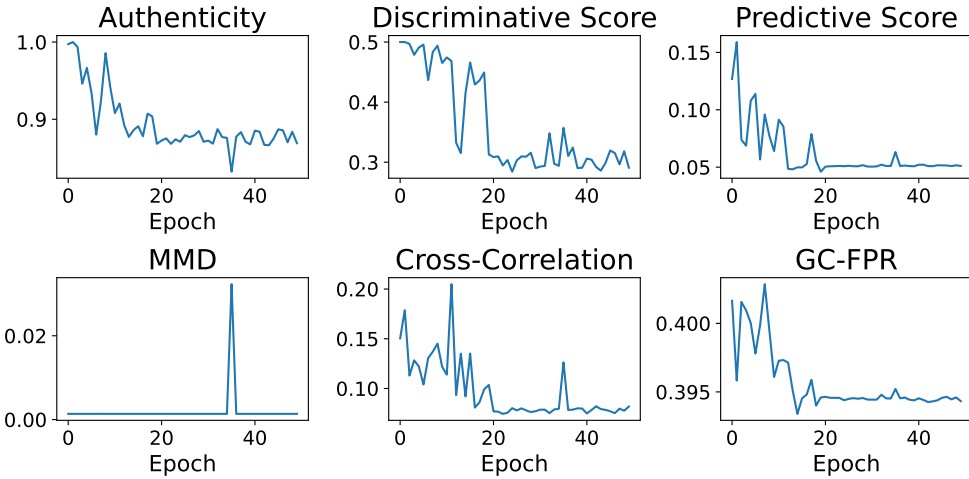

Figure 19: Evaluation metrics during training - AQI dataset.

## C.7 Graph Extraction Module

### C.7.1 Robustness of Graph Extraction

In this section, we report an additional experiment to highlight the robustness of the causal graphs to noise in the time-series. In particular, we trained `DiffCATS` in a setting where Gaussian noise $z \sim \mathcal{N}(0, \sigma^2)$ has been added to the training time-series. Table 10 reports the results of the Graph-FPR for $\sigma^2 = 0.005$ and $\sigma^2 = 0.01$ showing the robustness of the extraction to noise in the time-series.

Table 10: Graph-FPR with respect to noisy time-series.

| Dataset | Noise variance $(\sigma^2)$ | | |
|---|---|---|---|
| | 0 | 0.005 | 0.01 |
| Hénon | $0.04 \pm 0.00$ | $0.02 \pm 0.00$ | $0.02 \pm 0.00$ |
| Rivers | $0.07 \pm 0.00$ | $0.01 \pm 0.00$ | $0.01 \pm 0.00$ |

### C.7.2 Statistical test

Alternative approaches can be integrated to extract the causal graph based on the VAR coefficients. For example Hyvärinen et al. (2010) suggests conducting a statistical test of the significance of the coefficient. We tested the Dixon's Q Test to assess if there is at least one of the generated coefficients significantly higher than the others, resulting in a notable causal link. We evaluated the extracted causal graphs using this method with a $p$-value of 0.05. The Graph-FPR metric is 0.016 and 0.066 for the Rivers and Hénon datasets, respectively, which is very close to the score we obtained with the approach of Definition 1.

### C.7.3 Impact of $\rho$ and $q$

The results in Figure 20 show that the error in introducing edges remains contained when varying $q \in \{0.80, 0.85, 0.90, 0.95\}$ and $\rho \in \{0.001, 0.005, 0.01, 0.02\}$. As the parameter $\rho$ controlling the sparsity of the dataset increases, the Graph-FPR increases as well, meaning that the more relationships are kept in the dataset, the more it is the probability that they are wrong. However, it can be observed that the parameter $q$ is able to alleviate this by forcing each individual sample to keep only the coefficient of high magnitude.

We would like to highlight that the parameters $\rho$ and $q$ are used to extract causal graphs for the dataset from the generated coefficients. Critically, this means that a potential user of the model does not need to retrain the model to find the best $\rho$ and $q$ values.

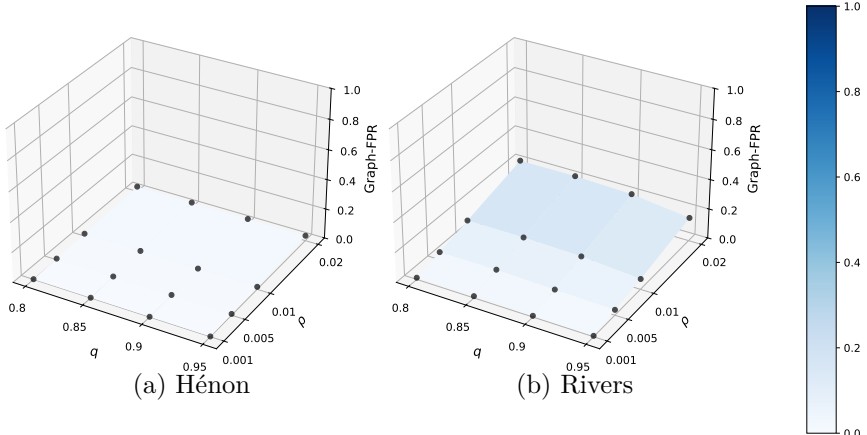

Figure 20: 3D surface plots showing Graph-FPR vs $q$ and $\rho$ for Hénon (a) and Rivers (b) datasets.

## C.8 Execution Time

We ran the experiments on a machine equipped with Intel Core i9-10920X CPU @ 3.50GHz, NVIDIA GeForce RTX 2060 GPU, and $8 \times 32$ GB DDR4 RAM.

A training phase of our model required $\sim$1 hour for 60 epochs. We have trained 6 variants of our model on 3 different datasets, for a total of $\sim$18 training hours.

In more detail, Table 11 shows the inference time of the models isolating the generation of the time-series and the extraction of the graph. It turns out that even if CausalTime is faster than DiffCATS in generating the time-series, the graph extraction through DeepSHAP introduces an important overload making it the slowest model.

Table 11: Inference time.

| Dataset | Model | | | | |
| | DiffCATS | DiffCATS with Graph | CausalTime | CausalTime with Graph | CR-VAE |
| --- | --- | --- | --- | --- | --- |
| Hénon | 1481ms | 1548ms | 465ms | 8790ms | 194ms |
| Rivers | 1425ms | 1492ms | 235ms | 4248ms | 148ms |
| AQI | 1395ms | 1535ms | 1349ms | 205s | 442ms |

## C.9 Increasing the Polynomial Degree

To accommodate non-linear causal links, the polynomial degree employed in the VAR reconstruction can be increased to approximate non-linear relationships. With a certain degree of approximation, any (non-linear) continuous function can be represented by a polynomial (Stone-Weierstrass Theorem), trading off approximation error of the real function with computational time, since there will be more coefficients generated by the model. As an example, we performed an experiment on the Hénon dataset with a polynomial of degree set to 2 since this dataset comprehends a quadratic relationship. Results shown in Table 12 show that the metrics to evaluate the time-series and the causal graph remain satisfactory even by raising the degree of the polynomial.

Table 12: Results on the Hénon dataset increasing the polynomial of the VAR.

| Metric | Polynomial Degree | |
| | 1 | 2 |
| --- | --- | --- |
| Discr. | 0.032 | 0.021 |
| Pred. | 0.156 | 0.226 |
| Graph-FPR | 0.017 | 0.019 |

## C.10 Causal Prediction Downstream Task involving Causal Graphs

In this section, we show the results of a downstream task involving the causal graphs. In particular, we considered the task of predicting the $i$-th feature of the multivariate time-series given its other dimensions and the corresponding causal graph.

Let $\boldsymbol{x}_{/i}$ be the $(d-1)$-variate time-series consisting of all the features but the $i$-th. The predictor model consists of a 2-layer LSTM to compute the embedding of the observed time-series and a linear layer $L_{graph}$ to learn the embedding of the associated graph $g$. The two embeddings are summed and passed through a linear layer ($L_{out}$) to output $\hat{\boldsymbol{x}}_i$. Formally, $\hat{\boldsymbol{x}}_i = L_{out}(e_{ts} + e_g)$ where $e_{ts} = \text{LSTM}(\boldsymbol{x}_{/i})$ and $e_g = L_{graph}(g)$. The model is trained to minimize the $\ell_1$-loss with respect to $\boldsymbol{x}_i$, i.e. the real $i$-th feature of $\boldsymbol{x}$.

We compared the above model with the case in which only the $(d-1)$-variate time-series $\boldsymbol{x}_{/i}$ is exploited to forecast $\boldsymbol{x}_i$, i.e. the model does not see the causal graph. The models are evaluated in terms of Mean Absolute Error (MAE) on an independent validation set. Table 13 shows the quantitative results highlighting that the causal graphs are useful to improve the reconstruction ability of the predictor, and Figure 21 shows some examples.

Table 13: Downstream task - Causal Prediction (Mean Absolute Error).

| Dataset | Predictor | |
|---|---|---|
| | w/ Causal Graph | w/o Causal Graph |
| Hénon | $\mathbf{0.13 \pm 0.01}$ | $0.19 \pm 0.01$ |
| Rivers | $\mathbf{0.01 \pm 0.00}$ | $0.02 \pm 0.00$ |

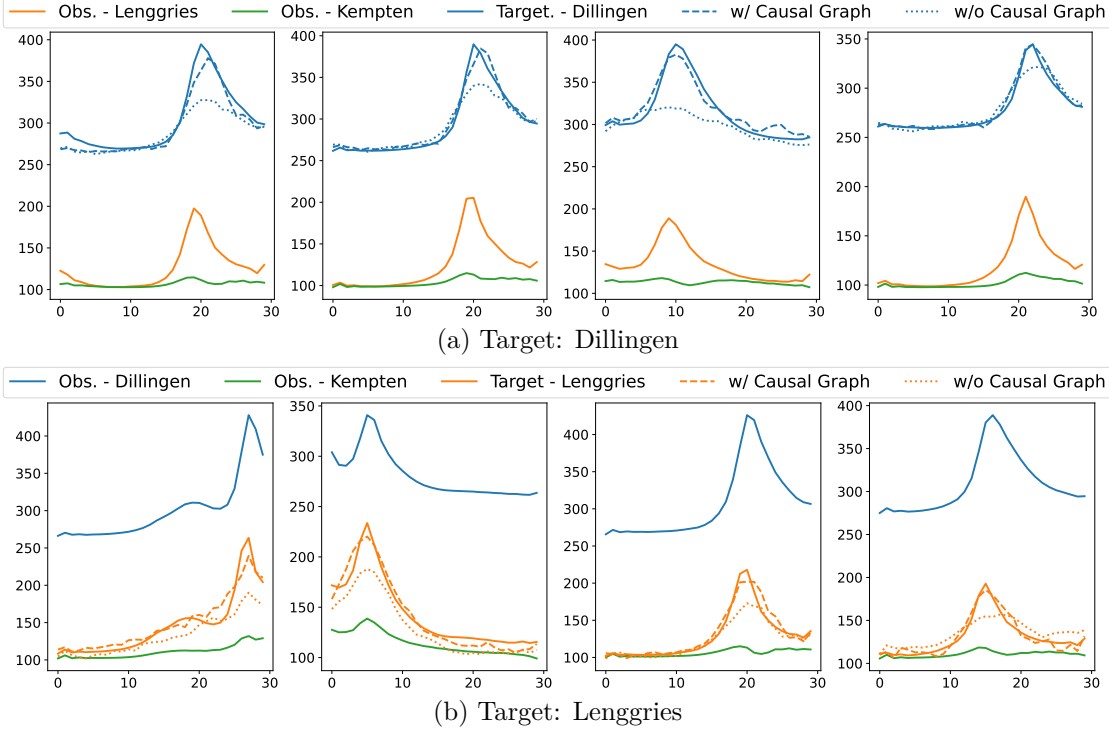

(a) Target: Dillingen

(b) Target: Lenggries

Figure 21: Causal Reconstruction.

## C.11 Causal Classification Downstream Task

Classifying time-series data based on their underlying causal dynamics represents a critical challenge in machine learning, especially because the causal structure of the system influences the observable behaviors. The broader significance of the causal classification task lies in its ability to enable meaningful grouping

of observations by their causal nature. This can be applied to problems like emergency detection, where identifying unsafe system dynamics relies on understanding the causal mechanisms driving those dynamics, or customer segmentation, where groups are formed based on inferred causal responses to interventions.

The `DiffCATS` framework supports this challenge by generating synthetic time-series explicitly paired with their associated causal graphs, enabling models to capture and leverage causal dependencies for classification tasks. Therefore, in this downstream task, the causal graph serves as a label or defining feature for the grouping of samples based on their underlying system dynamics, rather than being directly integrated into the prediction process as in the **Causal Predictive** task.

As a demonstration, we used the Rivers dataset. We identified the 10 most frequent causal graphs within the dataset and treated them as class labels; we also added an additional class to capture residual or less common causal dynamics. Each class represents a distinct causal graph that reflects different patterns in river discharge behavior. We then trained a 2-layer LSTM network on synthetic time series generated by `DiffCATS`, with the goal of predicting the underlying causal class (i.e., label) from the observed time-series. When evaluated on real time-series samples, the classifier achieved an F1-score of 0.69. This result highlights the potential of using synthetic datasets to enhance classification tasks by providing explicit structural labels.

To conclude, we believe `DiffCATS` provides a robust support for classification tasks by generating datasets with coupled time-series and causal graphs, ensuring high fidelity in both the signals and their causality. This capability is particularly valuable in scenarios where certain classes are underrepresented or entirely absent. In fact, our experiments demonstrate that the synthetic data produced by `DiffCATS` serve as effective surrogates for training accurate classifiers.

## D  Algorithms

We show the algorithm to reconstruct the whole time-series from the output of $\text{DEN}_\theta$ (i.e. the initial time-steps $\boldsymbol{x}_{start}$ and the set of coefficients $\boldsymbol{c}$) in Algorithm 1.

The sampling procedure of a synthetic couple $\langle \hat{\boldsymbol{x}}, \hat{g} \rangle$ is described in Algorithm 2.

---

**Algorithm 1** Reconstruction of $\hat{\boldsymbol{x}}$ from $\boldsymbol{x}_{start}$ and $\boldsymbol{c}$.

---

**Input:** $\boldsymbol{x}_{start}$, $\boldsymbol{c}$
**Output:** $\hat{\boldsymbol{x}}$
▷ $\boldsymbol{x}_{start}$.`shape = [`$d$`, `$\tau_{max}$`]`
▷ $\boldsymbol{c}$.`shape = [`$d$`, `$d \cdot \tau_{max}$`, `$L - \tau_{max}$`]`
$\hat{\boldsymbol{x}}_0 = \boldsymbol{x}_{start}$
**for** all $i$ from 0 to $L - \tau_{max}$ **do**
    `sup` $\leftarrow \hat{\boldsymbol{x}}_0$`[:,`$-\tau_{max}$`:].flatten()`
    $c \leftarrow \boldsymbol{c}$`[:,:,`$i$`]`
    $x \leftarrow$ `torch.einsum('a,ba->b', sup, c)`
    $\hat{\boldsymbol{x}}_0 \leftarrow$ `torch.cat([`$\hat{\boldsymbol{x}}_0$`,`$x$`.unsqueeze(`$-1$`)], dim=`$-1$`)`
**end for**
**Return** $\hat{\boldsymbol{x}}_0$

---

## E  TSCD Algorithms Benchmark

### E.1  Related work

Recent works have studied and tested causal discovery algorithms in several scenarios and domains. Hasan et al. (2023) provide a benchmark of 5 algorithms on both a synthetic and a real dataset, evaluating them using several binary classification metrics. Lawrence et al. (2021) use their framework to generate numerical datasets and evaluate 5 causal discovery algorithms, with an in-depth performance analysis concerning their diverse assumptions and hyper-parameters selection. Cheng et al. (2024) employs the synthetic version of

---

**Algorithm 2** Sampling of $\langle \hat{\boldsymbol{x}}, \hat{g} \rangle$.

---

**Input:** Trained denoising network $\mathrm{DEN}_\theta$
**Output:** $\hat{\boldsymbol{x}}, \hat{g}$
$\boldsymbol{x}_T \sim \mathcal{N}(\boldsymbol{0}, \mathbf{I})$
**for** all $t$ from $T$ to $0$ **do**
    $(\boldsymbol{x}_{\mathrm{start}}, \boldsymbol{c}) \leftarrow \mathrm{DEN}_\theta(\boldsymbol{x}_t, t)$
    $\hat{\boldsymbol{x}}_0 \leftarrow \mathrm{RECONSTRUCT}(\boldsymbol{x}_{\mathrm{start}}, \boldsymbol{c})$
    $\boldsymbol{x}_{t-1} \leftarrow \beta_t \cdot \frac{\sqrt{\hat{\alpha}_{t-1}}}{1 - \hat{\alpha}_t} \cdot \hat{\boldsymbol{x}}_0 + \frac{(1 - \hat{\alpha}_{t-1}) \cdot \sqrt{\alpha_t}}{1 - \hat{\alpha}_t} \cdot \boldsymbol{x}_t$
    **if** $t > 0$ **then**
        $\boldsymbol{x}_{t-1} \leftarrow \boldsymbol{x}_{t-1} + \beta_t \cdot \frac{1 - \hat{\alpha}_{t-1}}{1 - \hat{\alpha}_t} \cdot \boldsymbol{\epsilon}$
    **end if**
**end for**
$\hat{g} \leftarrow \mathrm{EXTRACTGRAPH}(\boldsymbol{c})$
**Return** $\hat{\boldsymbol{x}}_0, \hat{g}$

---

three real datasets to benchmark 13 representative state-of-the-art causal discovery algorithms. Finally, the very recent work of Li et al. (2024) incorporates LLMs to discover causal relationships from observational and interventional data. Their method is compared with 4 state-of-the-art baselines.

## E.2 Details on the algorithms

To evaluate the different TSCD algorithms we adapt/test them to our task using their source available code, whose repositories are listed below.

- GC: Granger Causality test implemented in the statsmodels[8] library.

- DYNOTEARS: `https://github.com/mckinsey/causalnex`

- NTS-NOTEARS: `https://github.com/xiangyu-sun-789/NTS-NOTEARS`

- PCMCI+: `https://github.com/jakobrunge/tigramite`

- Rhino: `https://github.com/microsoft/causica`

- CUTS / CUTS+: `https://github.com/jarrycyx/UNN`

- Neural-GC: `https://github.com/iancovert/Neural-GC`

- NGM: `https://github.com/alexisbellot/Graphical-modelling-continuous-time`

- LCCM: `https://github.com/edebrouwer/latentCCM`

- eSRU: `https://github.com/sakhanna/SRUforGCI`

- TCDF: `https://github.com/M-Nauta/TCDF`

The used hyper-parameters of the algorithms are reported in Table 14 (they are the same for all datasets).

## E.3 Benchmarking Causal Discovery Algorithms

Table 15 shows the results of our benchmark on a synthetic dataset where the causal graphs are extracted globally, following the procedure in Section 4.3.

---

[8]https://www.statsmodels.org/stable/index.html

Table 14: Hyper-parameters of the causal discovery algorithms.

| Algorithm | Hyper-parameter | Value |
|---|---|---|
| GC | `maxlag` | 2 |
| DYNOTEARS | `p`
`max_iter` | 2
100 |
| NTS-NOTEARS | `lags`
`w_threshold`
`h_tol` | 2
0.3
$1e-60$ |
| PCMCI+ | $\tau_{max}$
`PC`$_\alpha$ | 2
0.01 |
| Rhino | Noise Distribution
`init_rho`
`init_alpha` | Gaussian
30
0.2 |
| CUTS | Input step
$\lambda$
$\tau$ | 2
0.1
$0.1 \rightarrow 1$ |
| CUTS+ | Input step
$\lambda$
$\tau$ | 2
0.01
$0.1 \rightarrow 1$ |
| Neural-GC | Learning rate
$\lambda_{ridge}$
$\lambda$ | 0.05
0.01
$0.002 \rightarrow 0.02$ |
| NGM | Steps
Horizon
GL_reg | 500
5
0.1 |
| LCCM | `hidden_size`
Learning rate | 20
0.01 |
| eSRU | $\mu_1$
Learning rate
Batch size
Epochs | 1
0.005
30
500 |
| TCDF | $\tau$
Epochs
Learning rate | 10
1000
0.01 |

Table 15: Other results of the benchmark of Causal Discovery Algorithms. Bold and underline are used to highlight the best and the second best result, respectively.

| Method | AUROC |  |  | AUPRC |  |  |
|---|---|---|---|---|---|---|
|  | Hénon | Rivers | AQI | Hénon | Rivers | AQI |
| GC | $0.55 \pm 0.10$ | $0.73 \pm 0.16$ | $0.50 \pm 0.00$ | $0.45 \pm 0.11$ | $0.54 \pm 0.09$ | $0.48 \pm 0.08$ |
| DYNOTEARS | $0.45 \pm 0.11$ | $0.52 \pm 0.08$ | $0.50 \pm 0.00$ | $0.52 \pm 0.15$ | $0.56 \pm 0.08$ | $\underline{0.51 \pm 0.02}$ |
| NTS-NOTEARS | $0.64 \pm 0.14$ | $0.73 \pm 0.15$ | $0.50 \pm 0.00$ | $0.40 \pm 0.13$ | $0.55 \pm 0.14$ | $0.30 \pm 0.23$ |
| PCMCI+ | $\mathbf{0.84 \pm 0.08}$ | $\underline{0.82 \pm 0.08}$ | $\mathbf{0.68 \pm 0.00}$ | $0.54 \pm 0.09$ | $0.64 \pm 0.08$ | $0.50 \pm 0.03$ |
| Rhino | $0.50 \pm 0.02$ | $\underline{0.57 \pm 0.12}$ | $0.50 \pm 0.00$ | $0.52 \pm 0.01$ | $0.65 \pm 0.10$ | $0.51 \pm 0.03$ |
| CUTS | $0.81 \pm 0.10$ | $\mathbf{0.86 \pm 0.09}$ | $\underline{0.68 \pm 0.01}$ | $\underline{0.54 \pm 0.07}$ | $0.55 \pm 0.08$ | $\underline{0.51 \pm 0.02}$ |
| CUTS+ | $0.81 \pm 0.09$ | $0.75 \pm 0.09$ | $\underline{0.67 \pm 0.01}$ | $0.53 \pm 0.07$ | $0.58 \pm 0.08$ | $\underline{0.51 \pm 0.02}$ |
| Neural-GC | $0.67 \pm 0.00$ | $0.52 \pm 0.07$ | $0.50 \pm 0.01$ | $0.52 \pm 0.01$ | $0.53 \pm 0.05$ | $0.48 \pm 0.10$ |
| NGM | $\underline{0.84 \pm 0.13}$ | $0.80 \pm 0.13$ | $0.50 \pm 0.01$ | $\mathbf{0.63 \pm 0.16}$ | $\mathbf{0.81 \pm 0.12}$ | $0.47 \pm 0.13$ |
| LCCM | $\underline{0.50 \pm 0.00}$ | $0.50 \pm 0.00$ | $0.50 \pm 0.00$ | $0.51 \pm 0.00$ | $\underline{0.78 \pm 0.00}$ | $0.21 \pm 0.00$ |
| eSRU | $0.50 \pm 0.0$ | $0.71 \pm 0.10$ | $0.50 \pm 0.00$ | $0.53 \pm 0.01$ | $\underline{0.76 \pm 0.08}$ | $\mathbf{0.53 \pm 0.01}$ |
| TCDF | $0.50 \pm 0.0$ | $0.50 \pm 0.01$ | $0.50 \pm 0.00$ | $0.50 \pm 0.03$ | $0.53 \pm 0.09$ | $0.45 \pm 0.15$ |

## F  Limitations and Trade-off

Even if Table 1 reports the performance of strong generators (e.g., CSDI), our intended like-for-like comparison is against methods that can generate both (i) the multivariate time series and (ii) an associated causal graph, ideally with an explicit mechanism that promotes graph–sample consistency. `DiffCATS` is designed precisely for this paired generation setting: each synthetic sample is produced together with a corresponding causal graph, and the two are structurally coherent by construction. Instead, Base Diffusion and CSDI are optimized only for the fidelity of the synthetic time-series and are included as reference points, to show that `DiffCATS`' time-series quality is not far from high-performing time-series-only generators, while additionally providing

causal graphs. This is a trade-off induced by requiring causal-graph generation and consistency, rather than as an across-the-board weakness.

Furthermore, regarding efficiency, Table 1 shows `DiffCATS` inference around 1.4–1.5 s/sample, which is slower than CR-VAE (hundreds of ms), consistent with diffusion sampling requiring multiple denoising steps. At the same time, `DiffCATS` is substantially faster than CausalTime, whose inference is dominated by post-hoc feature-importance (DeepSHAP), reaching seconds to minutes in (e.g., up to 205s on AQI).

We also evaluated the number of GLOPs required by the models to generate a synthetic sample. The results for `DiffCATS` include all the denoising steps (100 in our case) while the results for CausalTime include all the steps involved in their autoregressive generation.

Table 16: GFLOPs per sample.

|  | DiffCATS | CR-VAE | CausalTime |
|---|---|---|---|
| Hénon | 14.885 | 0.004 | 6.613 |
| Rivers | 14.819 | 0.002 | 3.301 |
| AirQuality | 15.725 | 0.022 | 22.695 |

CR-VAE is consistently the cheapest (0.002–0.022 GFLOPs/sample) because it produces a sample in essentially a single lightweight forward pass of a variational autoencoder. `DiffCATS` requires 14.8–15.7 GFLOPs/sample across all three datasets due to its iterative diffusion sampling procedure. CausalTime's generation cost is competitive on small/moderate settings (3.301 GFLOPs on Rivers; 6.613 on Hénon), but increases markedly on the high-dimensional AirQuality setting (22.695 GFLOPs), consistent with the fact that its autoregressive generation must be repeated across the sequence and its per-step computation grows with dimensionality.

Importantly, the CausalTime GFLOPs reported here only count the forward computation needed to output a synthetic sample and do not include its post-processing for extracting the causal graph by interpreting the model with DeepSHAP. In practice, most of CausalTime's wall-clock inference overhead comes from this DeepSHAP-based attribution step, which requires many additional evaluations and can dominate runtime even when the raw GFLOPs/sample for generation looks moderate. By contrast, in our pipeline, even if `DiffCATS` requires more GFLOPs to generate a sample, the subsequent causal-graph extraction step is lightweight, so the end-to-end overhead is not driven by an expensive interpretability pass.

Finally, the GFLOPs trends across datasets also highlight scaling with the number of features (Rivers: 3, Hénon: 6, AirQuality: 36). `DiffCATS` stays roughly constant ($\sim$ 15 GFLOPs) across these experiments, while CR-VAE increases modestly (from 0.002 to 0.022 GFLOPs) and CausalTime grows substantially, especially at 36 features (22.695 GFLOPs). This suggests `DiffCATS` ' generation compute is comparatively less sensitive to feature dimensionality in our implementation, whereas CausalTime's autoregressive generator becomes significantly more expensive as dimensionality increases.

