# OpenReview forum: "DiffCATS: Causally Associated Time-Series Generation through Diffusion Models"
_TMLR — Accepted by TMLR_

### Review · Reviewer_FBxn · 2025-11-07

**Summary Of Contributions:**

This paper proposes DiffCATS, a diffusion model–based mechanism for generating time-series data that exhibit underlying causal structures. The method is designed to serve as a benchmark for evaluating time-series causal discovery (TSCD) algorithms. The proposed DiffCATS requires only multivariate time-series data for training, even when their causal structures are unknown, and it is capable of generating synthetic time-series samples associated with diverse causal graphs. Using datasets produced by several similar generative mechanisms, the authors apply multiple causal discovery algorithms to infer causal structures and evaluate both the inferred causal graphs and the generated time-series themselves through a variety of quantitative metrics, thereby demonstrating the usefulness of DiffCATS.

**Audience:**

Yes

**Audience Explanation:**

There is a consistently strong demand for causal discovery in complex real-world datasets, and thus, the development of benchmark data generation models to support such research is an essential and necessary contribution in this field.

**Claims And Evidence:**

No

**Claims Explanation:**

At least in its current form, the claims of this paper are not sufficiently supported for the following reasons:

1. The necessity of employing a diffusion model as the generative mechanism is not adequately discussed. The authors should clarify why other types of generative models are insufficient and emphasize what specific advantages or capabilities become possible *because* of using a diffusion model.
2. The definition of a causal relationship given in Definition 1 is crucial. In the final paragraph of Section 4, the paper asserts in words that this definition is consistent with existing formulations; however, this is a claim that warrants a formal, mathematical explanation, not merely a verbal statement.
3. As methods for causal discovery, LiNGAM and its variants occupy an important position in the field. It is unclear why these methods are neither included in the numerical comparisons nor cited in the related work section. The authors should justify this omission.

https://jmlr.org/papers/v7/shimizu06a.html
https://sites.google.com/view/sshimizu06/lingam

**Requested Changes:**

In addition to the three issues mentioned above, I would also request the following revision:
- In Appendices C.5 and C.6, please do more than simply present the results in graphical form. Include an explanation of what the graphs imply and what key observations or insights can be drawn from them.

---

> ### Author Response · Authors · 2025-12-20
>
> We thank the reviewer for their feedback. It follows our responses to reviewer's comments one by one.
> ## Why diffusion models?
> >The necessity of employing a diffusion model as the generative mechanism is not adequately discussed. The authors should clarify why other types of generative models are insufficient and emphasize what specific advantages or capabilities become possible because of using a diffusion model.
>
> We thank the reviewer for this question, we have now added a discussion of our modeling choices in the revised paper. We agree that our main contribution is a general methodology that could, in principle, be adapted to other generative models such as VAEs or GANs. However, our model choice is motivated by the current state of the art in time-series generation: VAE and GAN models often underperform diffusion models due to more restrictive assumptions and less stable training dynamics.
>
> Motivated by prior work, we therefore focus on a diffusion model for our framework, which we carefully design to our task. Moreover, beyond empirical performance, diffusion models offer additional flexibility that is particularly relevant for future extensions, including the possibility of guidance or constrained generation at each denoising step [1], which can be used to steer generation toward a target causal graph. %This is especially important when training data do not cover all possible causal structures.
>
> These properties make diffusion models particularly well suited to our problem and motivated our modeling choice. Moreover, given that diffusion models achieve state-of-the-art performance in time-series generation, we prioritize the evaluation of our method along other dimensions, rather than providing an additional benchmarking against alternative generative models. %, which is already covered in the literature.
>
> [1] Coletta, A., Gopalakrishnan, S., Borrajo, D., \& Vyetrenko, S. (2023). On the constrained time-series generation problem. Advances in Neural Information Processing Systems, 36, 61048-61059.
>
> ## Definition of causality
> > The definition of a causal relationship given in Definition 1 is crucial. In the final paragraph of Section 4, the paper asserts in words that this definition is consistent with existing formulations; however, this is a claim that warrants a formal, mathematical explanation, not merely a verbal statement.
>
> We apologize for the lack of clarity, and we thank the reviewer for highlighting this. We agree that additional details are needed to clarify how our formulation relates to established notions such as Granger causality (GC) and Transfer Entropy (TE). In the revised manuscript, we added in Appendix~A.2 a discussion that provides a clearer intuition behind our statement, divided into two points:
> - Connection to Granger causality
> - GC--TE equivalence under Gaussianity
>
> ## LiNGAM
> > As methods for causal discovery, LiNGAM and its variants occupy an important position in the field. It is unclear why these methods are neither included in the numerical comparisons nor cited in the related work section. The authors should justify this omission.
>
> We really thank the reviewer for pointing this out. We run several variants of the LiNGAM method and the results are  included and commented in the revised version of the paper.
>
> In general, LiNGAM-based approaches struggled with the non-linear Hénon dataset, but ICA-LiNGAM and DirectLiNGAM demonstrated competitive performance on the real-world datasets (Rivers and AQI), particularly in terms of AUROC.
>
> ## Comments in Appendices C.5 and C.6
> > In Appendices C.5 and C.6, please do more than simply present the results in graphical form. Include an explanation of what the graphs imply and what key observations or insights can be drawn from them.
>
> We thank the reviewer for pointing out the need for deeper interpretation of the results in the Appendices.
> We agree that simply presenting the graphs was insufficient to convey their significance.
>
> In the revised manuscript, we have expanded Appendix C.5 to explicitly discuss how the Kernel Density Estimation plots demonstrate DiffCATS' superior ability to capture marginal distributions and peak densities compared to baselines like CR-VAE.
>
> Furthermore, we have updated Appendix C.6 to interpret the training trajectories.
> We now highlight the rapid convergence of fidelity metrics (MMD, Discriminative Score) and explain the significance of the Authenticity trends, verifying that our model balances realism with novelty (avoiding memorization).
> We also discuss how the simultaneous improvement of metrics related to the time-series and metrics related to the causal graphs validates our joint training strategy.

---

> > ### Comment · Reviewer_FBxn · 2025-12-22
> >
> > Thank you for your response to the review report and for updating the manuscript.
> >
> > In particular, the discussion in Appendix A.2 regarding the relationship between Granger causality and TE will, I believe, greatly assist in understanding the positioning of the causality addressed in this study. Furthermore, I believe your responses to the comparison with LiNGAM-based methods and the rationale for using the diffusion model effectively address the reviewer's questions and concerns.

---

### Review · Reviewer_nzWU · 2025-11-10

**Summary Of Contributions:**

The authors present a method that is based on diffusion models to generate time series data, alongside a corresponding causal graph related to each generated sequence. The method is trained on real data, without relying on in the specification of a ground-truth graph, but rather infers a graph that best allows for the causal modelling of the seen real data.

The authors then go on to provide experimental evidence that their generated synthetic data has merit for a number of different tasks. First, they consider three time series datasets, of which the generate synthetic samples and calculate a variety of metrics for time series generation. The proposed method performs favourably or on par with competing approaches across the board. As a second task, the authors compare a large number of causal discovery algorithms on real data and synthetic data generated by their method. In the majority of settings, models achieve a similar performance on the synthetic data as they do on real data. The authors argue that this indicates the usefulness of the synthetic data their model generates for causal discovery. As a final task, the authors show the benefit of including the causal graph generated by their method in a causal prediction and causal classification task, respectively.

**Additional Comments:**

I cannot fully follow the motivation for generating synthetic time series for causal discovery in the way that the author's proposed method does. The concern does not seem to be overcoming artefacts of purely synthetic data such as sortability phenomena, as this is not mentioned in the manuscript. What is the benefit of generating this synthetic data? As the authors state, the goal is not to implicitly do causal discovery by means of generating data. Is the underlying motivation perhaps to have more data dor the development of causal discovery algorithms? If so, what is the benefit? Does the real data that is used to train the proposed method not suffice for this?

In my view, the criteria for acceptance to TMLR are met, but I would still be very interested in the authors' input on this topic of underlying motivation

**Audience:**

Yes

**Audience Explanation:**

The authors reference prior works which they extend, therefore I would consider the findings in this paper to be interesting to the audience interested in these works, as well.

**Claims And Evidence:**

Yes

**Claims Explanation:**

While the motivation for some of the experiments and claims is not crystal clear to me (see my additional comments below), the authors provide ample evidence for the claims they make in terms of experiments across many settings.

To the best of my judgement, there are no technical errors in this paper.

**Requested Changes:**

Please tone down the claim in the first sentence of the introduction (as well as in the abstract) that real-world phenomena have an *inherent* causal structure. The description of a relationships between variables is a modelling choice and should be described as such. Whether or not the world truly functions according to causal models is an entirely different question.

If you are using the biblatex package, please use the ``\citep`` command for citations, as the current style produces inline citations that are read as part of the text, rather than citations with clear parentheses to distinguishing them.

The caption for Figure 3 would benefit from being more verbose. Currently, I cannot understand the figure of the model from only reading the caption.

---

> ### Author Response · Authors · 2025-12-20
>
> We thank the reviewer for their feedback. It follows our responses to reviewer's comments one by one.
>
> ## Causality as a modelling choice
> > Please tone down the claim in the first sentence of the introduction (as well as in the abstract) that real-world phenomena have an inherent causal structure. The description of a relationships between variables is a modelling choice and should be described as such. Whether or not the world truly functions according to causal models is an entirely different question.
>
> We thank the reviewer for spotting this imprecision and for the helpful clarification. We have now revised such statements.
>
> ## Cite format
> >If you are using the biblatex package, please use the \textbackslash citep command for citations, as the current style produces inline citations that are read as part of the text, rather than citations with clear parentheses to distinguishing them.
>
> We thank the reviewer for highlighting this. We fixed the issue with citations, improving the presentation of our work.
>
> ## Caption of Figure 3
> >The caption for Figure 3 would benefit from being more verbose. Currently, I cannot understand the figure of the model from only reading the caption.
>
> We thank the reviewer for this point. We have now improved the caption of Figure 3 to make it self-contained.
>
> ## Why synthetic data?
> > I cannot fully follow the motivation for generating synthetic time series for causal discovery in the way that the author's proposed method does. The concern does not seem to be overcoming artefacts of purely synthetic data such as sortability phenomena, as this is not mentioned in the manuscript. What is the benefit of generating this synthetic data? As the authors state, the goal is not to implicitly do causal discovery by means of generating data. Is the underlying motivation perhaps to have more data dor the development of causal discovery algorithms? If so, what is the benefit? Does the real data that is used to train the proposed method not suffice for this?
>
> We thank the reviewer for the question. As correctly noted, the main motivation of our approach is to support the development of causal discovery algorithms using synthetic data to increase data volume, but also data diversity and variety, which are crucial for robust development and evaluation of causal discovery methods.
>
> In fact, while real-world data can be sufficient in some cases, they are often limited in size or quality, typically lack known ground-truth causal structure, or exhibit low variability even when such structure is available. These limitations may hinder the development and evaluation of causal discovery algorithms.
> Our synthetic data can complement real data by providing known ground truth and diverse causal structures, enabling more controlled development and experimentation. Prior work has already highlighted the importance of synthetic data for causal discovery [1], and, in general, synthetic data has proven valuable across several domains for mitigating data scarcity and bias [2].
>
> Concretely, our synthetic datasets enable systematic assessment of causal discovery methods under controlled conditions, support counterfactual analysis, and allow evaluation across diverse causal regimes, as demonstrated in our experiments and downstream tasks. In addition, our data supports the training of data-hungry models (e.g., deep learning approaches such as Rhino) by providing sufficient volume and diversity to improve generalization.
>
> Finally, while we acknowledge the limitations of purely synthetic data, we argue that our data provides a complementary resource to support the development and evaluation of novel causal discovery algorithms. Moreover, we believe our approach may foster future research in this area.
>
> [1] - Yuxiao Cheng, Ziqian Wang, Tingxiong Xiao, Qin Zhong, Jinli Suo, and Kunlun He. Causaltime: Realistically generated time-series for benchmarking of causal discovery. In The Twelfth International Conference on Learning Representations, 2024.
>
> [2] - Jordon, James, et al. "Synthetic Data--what, why and how?." arXiv preprint arXiv:2205.03257 (2022).

---

> > ### Comment · Reviewer_nzWU · 2025-12-22
> > **Response to the authors**
> >
> > I thank the authors for their revisions and clarifications provided by their response.
> >
> > I believe the proposed revisions clarify the message of the paper further and vote for acceptance.

---

### Review · Reviewer_eBEv · 2025-12-14

**Summary Of Contributions:**

The paper introduces DiffCATS, a novel generative framework based on diffusion models. DiffCATS simultaneously generates a multivariate time-series and its corresponding ground-truth causal graph, requiring only observational data for training. The core mechanism uses a Vector Autoregressive structure within the diffusion process, where the causal graph is directly and transparently derived from the learned VAR coefficients. The utility of the generated data is demonstrated in benchmarking TSCD algorithms, causal prediction, and causal classification.

Strengths: 1. The model ensures structural consistency between the generated time-series and its causal graph, which is a key objective for realistic synthetic data generation. 2. By embedding the causal structure directly through VAR coefficients, the resulting graph is naturally interpretable.

Weaknesses: 1. The definition for extracting the causal graph relies on two critical hyperparameters, $\rho$ (percentage of relationships) and $q$ (quantile threshold). I'm wondering how the two parameters affect the final performance. 2. The comparison in Table 1 shows that the proposed method still lags behind previous models in a couple of cases. Can the authors further discuss it, including the capability boundary of DiffCATS? I also notice that the inference time of DiffCATS is also somewhat long. More discussion about the balance of performance and efficiency is also needed here. 3. I find many references are not corrected cited in the paper. For example, in the first paragraph, all the citations should be in the format like (Hasan et al., 2023) rather than the current Hasan et al. (2023) because they are not a part of the sentence structure. Most citations in this paper have this problem.

**Audience:**

Yes

**Audience Explanation:**

1. The paper directly addresses a critical and widely recognized bottleneck in Time-Series Causal Discovery research. The researchers in this field will find the proposed method helpful in their study.

2. The paper presents a novel application of diffusion models to the multivariate time-series domain, which is of clear interest to generative model researchers.

**Claims And Evidence:**

Yes

**Claims Explanation:**

1. The core claims of superior realism and causal graph fidelity are backed by comparisons against multiple state-of-the-art baselines on three datasets. The paper reports superior or comparable scores on standard metrics like MMD and Discriminative Score, validating the realism of the generated data. The achievement of the best GC-FPR and Graph-FPR on all three datasets is a convincing piece of evidence that the generated graphs minimize false causal links.

2. The synthetic data served as a viable testbed for benchmarking several widely used TSCD algorithms, which also supports their claim of practical utility.

**Requested Changes:**

1. Provide a sensitivity analysis and discussion on the impact of the graph extraction hyperparameters $\rho$ and $q$.

2. Supplement more discussons about some scenarios that the proposed method does not perform as well as the previous models, as well as a balance between performance and efficiency.

3. For generative models, it will be very helpful if the authors can include the model size or number of flops for DiffCATS and baselines in the comparison tables.

4. Correct the wrong citation formats.

---

> ### Author Response · Authors · 2025-12-20
>
> We thank the reviewer for their feedback. It follows our responses to the reviewer's comments one by one.
> ## Impact of $\rho$ and $q$
> We thank the reviewer for pointing this out.
> We evaluated the Graph-FPR metric by varying the parameters $\rho$ and $q$.
> The results in Figure 20 in Appendix C.7.3 show that the error in introducing edges remains contained when varying $q \in \{0.80, 0.85, 0.90,0.95\}$ and $\rho \in \{0.001, 0.005, 0.01,0.02\}$.
>
> As the parameter $\rho$ controlling the sparsity of the dataset increases, the Graph-FPR increases as well, meaning that the more relationships are kept in the dataset, the more it is the probability that they are wrong.
> However, it can be observed that the parameter $q$ is able to alleviate this by forcing each individual sample to keep only the coefficient of high magnitude.
>
> We would like to highlight that the parameters $\rho$ and $q$ are used to extract causal graphs for the dataset from the generated coefficients.
> Critically, this means that a potential user of the model does not need to retrain the model to find the best $\rho$ and $q$ values.
> ## Limitations and Trade-off
> We thank the reviewer for this comment. We have revised the limitation section of the manuscript and added the Appendix F section to include the evaluation scope and the observed trade-offs explicitly.
>
> Our intended like-for-like comparison is against methods that can generate both (i) the multivariate time series and (ii) an associated causal graph, ideally with an explicit mechanism that promotes graph–sample consistency.
> In Table 1, we report the performance of strong generators (e.g., CSDI), optimizing time-series fidelity but not able to generate causal graphs (nor provide a mechanism to ensure consistency between a generated series and any graph).
> We clarified this in the text, pointing out that Base Diffusion and CSDI are included as reference points, to show that DiffCATS’ time-series quality is not far from high-performing time-series-only generators, while additionally providing causal graphs.
>
> Furthermore, regarding efficiency, Table 1 shows DiffCATS inference around ~1.4–1.5 s/sample, which is slower than CR-VAE (hundreds of ms), consistent with diffusion sampling requiring multiple denoising steps.
> At the same time, DiffCATS is substantially faster than CausalTime, whose inference is dominated by post-hoc feature-importance (DeepSHAP), reaching seconds to minutes in Table 1 (e.g., up to 205 s on AQI).
> We added a short practitioner-oriented paragraph clarifying when one might prefer: CR-VAE (fastest, but learns a fixed causal pattern rather than per-sample graphs), time-series-only diffusion models, like CSDI (if graphs are not needed), and DiffCATS (if the user needs coherent per-sample causal graphs without expensive post-hoc explanation).
>
> Regarding the capability boundary discussion, DiffCATS reconstructs samples via a linear VAR mechanism, hence generated causal effects are linear under the current formulation; in the appendix we propose an extension path (e.g., increasing polynomial degree) for non-linear links.
> We also added an explicit note such as dependence on the chosen maximum lag $\tau_{\text{max}}$ (which bounds representable temporal dependencies).
>
> We also evaluated the number of GLOPs required by the models to generate a synthetic sample (Table 16 in Appendix F).
> The results for DiffCATS include all the denoising steps (100 in our case) while the results for CausalTime include all the steps involved in their autoregressive generation.
>
> CR-VAE is consistently the cheapest (0.002–0.022 GFLOPs/sample) because it produces a sample in essentially a single lightweight forward pass of a variational autoencoder.
> Importantly, the CausalTime GFLOPs reported here only count the forward computation needed to output a synthetic sample and do not include its post-processing for extracting the causal graph by interpreting the model with DeepSHAP.
> In practice, most of CausalTime’s wall-clock inference overhead comes from this DeepSHAP-based attribution step, which requires many additional evaluations and can dominate runtime even when the raw GFLOPs/sample for generation looks moderate.
> By contrast, in our pipeline, even when DiffCATS expends more GFLOPs to generate a sample, the subsequent causal-graph extraction step is lightweight, so the end-to-end overhead is not driven by an expensive interpretability pass.
>
> Finally, DiffCATS stays roughly constant ($\sim15$ GFLOPs) across datasets with different number of features, while CR-VAE increases modestly (from 0.002 to 0.022 GFLOPs) and CausalTime grows substantially, especially at 36 features (22.695 GFLOPs).

---

> > ### Comment · Reviewer_eBEv · 2026-01-07
> >
> > Thank the authors for the detailed response and paper revision. My concerns are cleared.

---

### Comment · Action_Editor_QPsS · 2025-12-14
**Start of author discussion**

Dear all,
the author's discussion started

---

### Decision · Action_Editor_QPsS · 2026-01-16

**Recommendation:** Accept as is

**Audience:**

Yes

**Audience Explanation:**

Indeed, the paper has severla strong points as
A) The model ensures structural consistency between the generated time-series and its causal graph, which is a key objective for realistic synthetic data generation.
B) the idea to embed the causal structure directly through VAR coefficients gives interpretability.
C) As FBxn mentions, the proposed DiffCATS requires only multivariate time-series data for training, even when their causal structures are unknown, and it is capable of generating synthetic time-series samples associated with diverse causal graphs

**Claims And Evidence:**

Yes

**Claims Explanation:**

All the reviewers, also FBxn who first raised some issues, agree that the claims made by the paper are sensibly supported by the results of numerical experiments